# An ARF GTPase module promoting invasion and metastasis through regulating phosphoinositide metabolism

Marisa Nacke[1,2,4], Emma Sandilands[1,2,4], Konstantina Nikolatou[1,2,4], Álvaro Román-Fernández[1,2], Susan Mason[2], Rachana Patel[2], Sergio Lilla[2], Tamas Yelland[2], Laura C. A. Galbraith[2], Eva C. Freckmann[1,2], Lynn McGarry[2], Jennifer P. Morton[1,2], Emma Shanks[2], Hing Y. Leung[1,2], Elke Markert[2], Shehab Ismail[1,2,3], Sara Zanivan[1,2], Karen Blyth[1,2] & David M. Bryant[1,2 ✉]

The signalling pathways underpinning cell growth and invasion use overlapping components, yet how mutually exclusive cellular responses occur is unclear. Here, we report development of 3-Dimensional culture analyses to separately quantify growth and invasion. We identify that alternate variants of IQSEC1, an ARF GTPase Exchange Factor, act as switches to promote invasion over growth by controlling phosphoinositide metabolism. All IQSEC1 variants activate ARF5- and ARF6-dependent PIP5-kinase to promote PI(3,4,5)P$_3$-AKT signalling and growth. In contrast, select pro-invasive IQSEC1 variants promote PI(3,4,5)P$_3$ production to form invasion-driving protrusions. Inhibition of IQSEC1 attenuates invasion in vitro and metastasis in vivo. Induction of pro-invasive IQSEC1 variants and elevated *IQSEC1* expression occurs in a number of tumour types and is associated with higher-grade metastatic cancer, activation of PI(3,4,5)P$_3$ signalling, and predicts long-term poor outcome across multiple cancers. IQSEC1-regulated phosphoinositide metabolism therefore is a switch to induce invasion over growth in response to the same external signal. Targeting IQSEC1 as the central regulator of this switch may represent a therapeutic vulnerability to stop metastasis.

---

[1] Institute of Cancer Sciences, University of Glasgow, Glasgow, UK. [2] The CRUK Beatson Institute, Glasgow, UK. [3] Present address: Department of Chemistry, KU Leuven, Celestijnenlaan, Belgium. [4] These authors contributed equally: Marisa Nacke, Emma Sandilands, Konstantina Nikolatou. ✉email: david.bryant@glasgow.ac.uk

A central conundrum in biology is that highly overlapping signalling pathways regulate distinct biological outputs. For instance, activation of receptor tyrosine kinases (RTKs), and their downstream effectors such as the MAPK-ERK and PI3K-Akt pathway, can induce either growth or invasion. Since the development of invasive features and the progressive loss of normal tissue organisation are hallmarks of tumour progression[1] understanding how cells decode their response to external signals has significant clinical implications.

Metastasis is the major cause of cancer-related death, increasingly recognised as a collective migration event. For a collection of epithelial cells to invade they must undergo rearrangement of normal apical–basal cellular polarisation to clusters or chains of cells lead by an invasive front. The use of 3-dimensional (3D) culture systems, whereby epithelial cells are embedded in gels of extracellular matrix (ECM) to undergo collective morphogenesis, has illuminated the molecular mechanisms of collective cell polarisation[2–6]. Such polarity rearrangements can be achieved via altered membrane trafficking of morphogenesis-regulating proteins. For collective invasion to occur, signalling receptors, such as RTKs, must be directed to domains where a pro-invasive ligand is exposed. Some RTKs, such as the HGF receptor Met, require internalisation and endosomal localisation for full oncogenic signalling[7–9]. Additional membrane trafficking steps, such as recycling back to the cortex, can provide sustained signalling to effectors[10]. However, what controls whether such signals result in tumour cell growth versus the induction of collective invasion and metastasis remains unclear.

The ARF family of small GTPases (ARF1–6) are implicated in the membrane trafficking and signalling mechanism underpinning single cell invasion[11–16]. ARF GTPases have fundamental roles in vesicular transport by regulating assembly of coat complexes, lipid-modifying enzymes, and recruiting regulators of other GTPases onto membranes undergoing scission[17]. ARF6 in particular controls internalisation and recycling of RTKs by acting in concert with a number of GTP exchange factors (GEFs)[18–21] and GTPase-activating proteins (GAPs)[22–26]. Accordingly, small molecule ARF GEF or GAP inhibitors have been used to control invasion in vitro and metastasis in vivo[23,27–29].

Despite the core requirement for ARF GTPases in membrane trafficking steps controlling RTK signalling, it remains unclear how they could promote an invasion response, rather than growth, from an RTK. Are particular ARF-regulated trafficking pathways induced in invasive cells, such as to enhance endocytic recycling and sustained RTK activation, to promote invasion over growth? Here, we describe that specific pro-invasive transcript variants of the ARF GEF IQSEC1 are upregulated in invasive tumours. These alternate variant proteins act as scaffolds to direct phosphoinositide metabolism to induce invasive protrusions. We identify that IQSEC1 can be targeted to inhibit collective invasion in vitro and metastasis in vivo.

## Results

### Expression of the ARF GEF IQSEC1 is associated with poor clinical outcome.
We examined whether ARF GTPase expression was associated with the acquisition of invasive behaviours. We used 3D culture of prostate cancer cell lines to represent the transition from non-tumorigenic to highly metastatic (Fig. 1a). Non-tumorigenic RWPE-1 cells[30] formed acini with a central lumen (Fig. 1a, b). RWPE-2, an oncogenic KRAS-expressing RWPE-1 variant[30] formed lumen-lacking aggregates, some of which developed invasive cell chains (Fig. 1b, white arrowheads). Bone metastasis-derived PC3 cells[31] grew as heterogeneous acini, variably forming round, locally spread, or spindle-shaped invasive cell chains (Fig. 1c). Multiday live imaging revealed that invasive

chains were derived from single cells that first formed spherical aggregates (growth phase) that initially protruded into matrix, then elongated into chains (Fig. 1c, arrowheads, invasion phase).

ARF GTPases function with GEFs and GAPs that control their nucleotide-association state (Fig. 1d)[17]. We observed an association of the GEF IQSEC1 (also called BRAG2/GEP100[32]) with prostate tumorigenesis. All ARF GTPases, and a number of ARF GEFs were upregulated in PC3 cells (Fig. 1e, g). IQSEC1, including multiple isoforms, was a strongly expressed GEF in RWPE-2 and PC3. Normal prostate and breast cells expressed low levels of IQSEC1, while invasive lines (PC3, VCaP) had upregulated IQSEC1 mRNA, with little variation in copy number (Fig. 1g). Western blotting of androgen receptor (AR)-proficient and -deficient prostate lines confirmed upregulation of ARF6 and IQSEC1 protein expression in metastatic prostate cancer cell lines (LNCaP, VCaP, DU145, PC3) (Fig. 1h). We focused on dissecting IQSEC1 molecular function.

### IQSEC1 is a regulator of collective cell invasion.
We examined the contribution of IQSEC1 to cell growth and movement. Publicly available IQSEC1 transcript information revealed multiple variants occurring through combinatorial use of alternate translational initiation sites and alternate splicing (Fig. 2a; Supplementary Table 1). Western blotting suggested simultaneous expression of multiple variants in PC3 cells, with three IQSEC1 bands depleted by IQSEC1-specific shRNAs (Supplementary Fig. 1a). IQSEC1 depletion reduced proliferation proportional to knockdown efficiency (Supplementary Fig. 1a, b). As PC3 cells grow as a mixed morphology 3D culture (Fig. 1c), we developed a machine learning approach to determine whether this heterogeneity also occurred in 2D (Supplementary Fig. 1c). Mirroring 3D collective phenotypes (Fig. 1c), single PC3 cells in 2D culture could be classified into round (54%), spread (21%) and spindle phenotypes (17%) (Supplementary Fig. 1d, e). IQSEC1 depletion selectively abolished spindle characteristics, causing increased spread behaviours (Supplementary Fig. 1f).

We examined whether IQSEC1-dependent spindle shape was required for cell movement. Live imaging of wounded 2D monolayers revealed cells of various shapes rapidly move into the wound (Supplementary Fig. 1g). IQSEC1-depleted cells displayed a modest defect in migration. In contrast, in 3D invasion assays wounded monolayers embedded in ECM (Matrigel) invade by forming spindle-shaped protrusions that develop into multicellular chains (Supplementary Fig. 1h, white arrowheads). In IQSEC1-depleted cells, although some protrusions formed, multicellular chain formation and invasive activity was strongly compromised (Supplementary Fig. 1h, i). Thus, IQSEC1 is required for growth and multicellular invasive chain formation, resulting in defects in morphogenesis where cell elongation is required (Supplementary Fig. 1j).

### IQSEC1 is a regulator of collective 3D invasion.
We dissected IQSEC1 contribution to growth and/or invasion in 3D PC3 acini, which can form without (round) or with (spindle) invasive characteristics (Fig. 1c). In order to quantify this growth versus invasive behaviour we developed an automated method of quantitation from hourly imaging of hundreds to thousands of 3D acini over multiple days (see the "Methods" section and Supplementary Fig. 2a).

Elevated IQSEC1 expression in PC3 cells, compared to RWPE-1 cells, was maintained upon plating of both cell types into 3D culture (Fig. 2b). Growth of acini from single cells could be measured by increased area over time (Supplementary Fig. 2b, c). The progressive development of protrusive invasion could be determined using a 'Compactness' measurement

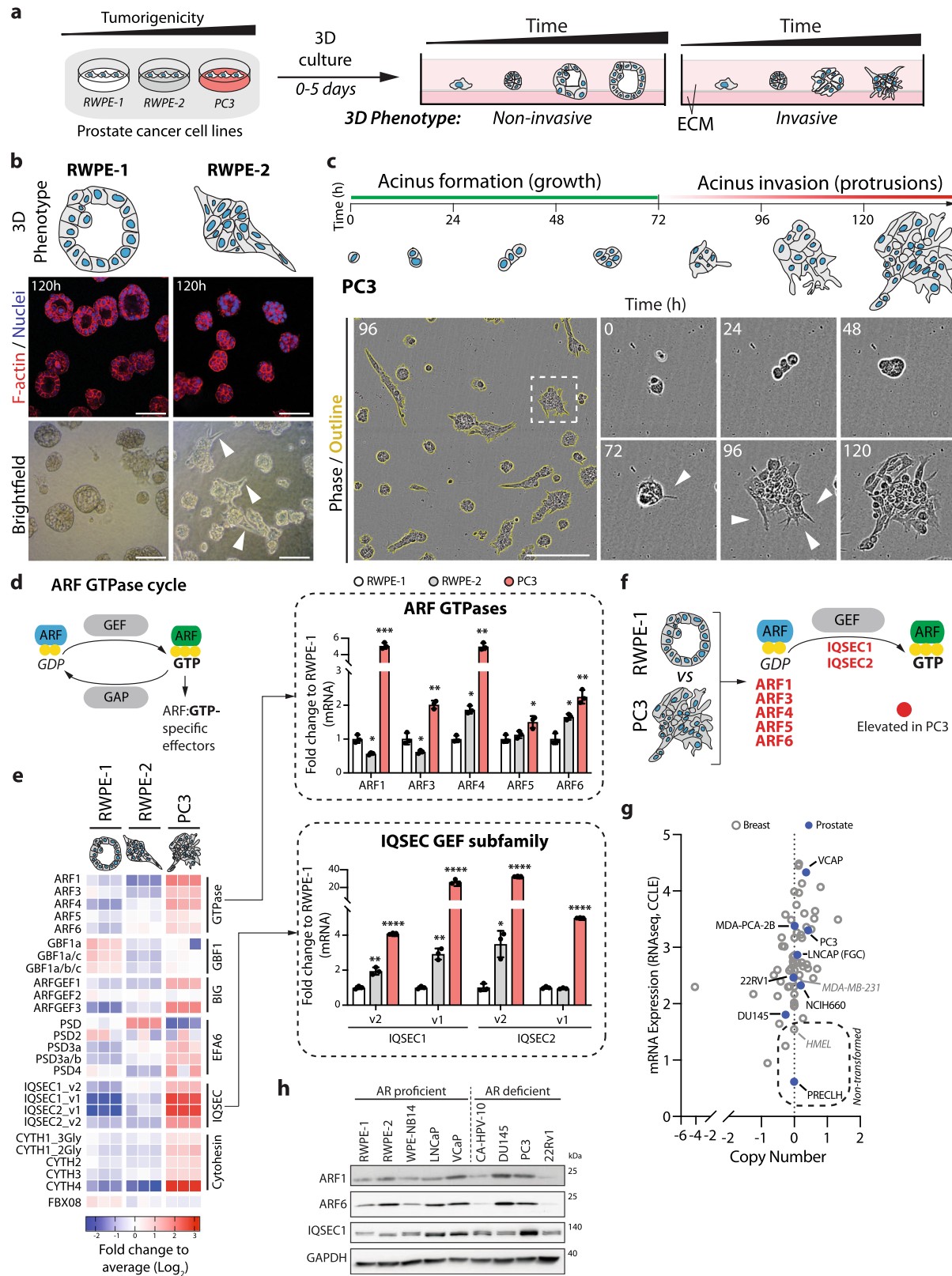

(Supplementary Fig. 2b, c; Supplementary Movies 1 and 2). Similar to 2D, IQSEC1 depletion modestly decreased 3D acinar growth (Supplementary Fig. 2c), proportional to depletion levels (Supplementary Fig. 1a). Mirroring 3D wound invasion defects (Supplementary Fig. 1h), the most prominent effect of IQSEC1 depletion was abolished protrusive invasion (Supplementary Fig. 2b, c).

We examined whether protrusion-forming activity was a common feature of all IQSEC1 isoforms. We focused on the four annotated variants (v1–4; Fig. 2a). IQSEC1 v1–4 possess the IQSEC family-defining features of a calmodulin-binding IQ domain, a catalytic SEC7 ARF GEF domain, and a lipid-binding PH domain. Alternate initiation sites provide v1 or v2 with unique N-termini, followed by a common N-terminal extension.

**Fig. 1 Upregulation of IQSEC1 is associated with tumorigenesis. a** Schema, prostate cell lines forming non-invasive or invasive 3D acini in extracellular matrix (ECM). **b** Cartoon, phenotype of typical RWPE-1 and RWPE-2 acini. Confocal (F-actin (red) and nuclei (blue)) and brightfield images show RWPE-1 and RWPE-2 acini (120 h). Arrowheads, protrusions. $n = 3$ independent experiments with 3 replicates/condition. Scale bar, 20 μm. **c** Schema, PC3 acini form (grow) and invade (protrusions) through ECM over time. Phase contrast images of PC3 acini where higher magnification of boxed region at different time points is shown. Arrowheads, protrusions. $n = 3$ independent experiments with 3 replicates/condition. Scale bar, 100 μm. **d** Cartoon, ARF GTPase cycle. **e** Heatmap representation of mRNA expression from q-PCR. Data are normalised to RWPE-1 and presented as the log2-transformed fold change compared to the average of all values. Bar graphs summarise fold changes of *ARF* and *IQSEC* mRNA levels. Mean ± s.d., $n = 3$ biological replicates. *p*-values; Student's *t*-test (2-tailed). *$p \leq 0.05$, **$p \leq 0.01$, ***$p \leq 0.001$ and ****$p \leq 0.0001$. **f** Schema, elevated activation of ARF GTPases in PC3 cells by GEFs such as IQSEC1. **g** Graph generated using RNAseq data from the Cancer Cell Line Encyclopaedia (CCLE) comparing *IQSEC1* gene copy number and mRNA expression levels in multiple breast and prostate cancer and non-transformed cell lines. **h** Western blot analysis of androgen receptor (AR) proficient or deficient prostate cell lines using anti-ARF1, ARF6, pan-IQSEC1 isoform and GAPDH (as sample control) antibodies. $n = 2$ independent experiments.

Three alternate C-termini can occur, with the v4-type tail truncating a potential nuclear localisation sequence.

We stably, individually restored each RNAi-resistant GFP-tagged IQSEC1 variant in the background of depletion of all endogenous IQSEC1 transcripts. All IQSEC1 variants could restore growth and/or invasion to levels matching or above control cells (Fig. 2c–f). Variants containing the N-terminal extension (v1, v2) conferred the strongest effect on growth. In contrast, protrusive activity occurred in an isoform-selective manner: only v2 (also known as BRAG2b) restored spindle behaviours in both 2D and 3D to levels surpassing controls (Fig. 2e, f; S2D). To corroborate the live imaging approach we examined fixed acini through 3D confocal imaging. Restoration of v2 to IQSEC1 KD cells increased the total number of nuclei, the level of a proliferation marker (Ki67) and suppressed apoptosis without changing total acinus volume (Supplementary Fig. 2e–h). This represented formation of protrusions and disruption to lumen formation. In contrast, v4 failed to rescue IQSEC1 KD-induced growth defects, but instead increased area by increasing acinus volume without changing cell number, due to the presence of a lumen. Thus, IQSEC1 v2 is a major regulator of 3D growth and invasion.

We explored the IQSEC1 v2 properties that promote invasion. IQSEC1 can function in ARF GTPase-dependent endocytosis at the cell cortex, and in the nucleus to control nucleolar architecture[32,33]. V3 showed predominantly nuclear localisation, while v4 was cortical (Figs. 2g, S2i). V1 showed mixed cortical and cytoplasmic localisation. In contrast, v2 displayed a mix of cytoplasmic, cortical and nuclear labelling (Fig. 2g, green arrowheads, S2i). We reasoned that in overexpressed GFP-tagged variants may mask vesicular pools. Accordingly, anti-IQSEC1 antibodies directed to either all isoforms or to v2-specifically (Fig. 2h) labelled both the cytoplasm and tubulovesicular compartments behind the tips of invasive acinar protrusions in 3D (Fig. 2i–k, arrowheads, S2j, k). Thus, one locale of IQSEC1 v2 function is focally at endosomes in protrusion tips (Fig. 2i–k).

We mapped the domains of v2 responsible for its localisation and potent invasion-inducing activity. We expected that this would be conferred by the unique v2 N-terminus. However, using mutants and chimeras of IQSEC1 isoforms revealed a nuanced and combinatorial effect of alternate N- and C-termini on IQSEC1 localisation and protrusion induction (Supplementary Fig. 3a–g). Surprisingly, the unique v2 N-terminus (2N) was not required for invasion. Any variant containing the v2-tail and any N-terminal extension promoted invasion. Accordingly, replacing the unique v2 N-terminus (2N) with that from v1 (1N) even enhanced invasion over and above that of v2 (Supplementary Fig. 3e–g). This was concomitant with a shift away from nuclear or cortical localisation, towards the cytoplasm (Supplementary Fig. 3c, 1N/2C chimera). This is in contrast to expression of v1 which did not induce invasion (Fig. 2f). This is due to the

presence of a v1-type C-terminus (1C) which in all experiments inhibits invasion, concomitant with a more general cortical recruitment. A v2 tail without N-terminal extension (i.e. v3) similarly does not promote invasion. This maps the invasion-inducing activity of IQSEC1 to the N-terminal extension that is common to only v1 and v2. However, the type of C-terminus influences whether this N-terminus induces invasion (Supplementary Fig. 3a–g). Mutational inactivation of GEF activity (IQSEC1 v2$^{GEF*}$) confirmed that IQSEC1 v2-induced spindle shape in 2D (Supplementary Fig. 3b) and protrusive invasion in 3D (Supplementary Fig. 3d–g) are due to its ability to activate ARF GTPases. Thus, the N-terminal common extension in v2 confers enhanced invasive activity in a GEF-dependent manner.

## IQSEC1 activates ARF5/6 in distinct locations within protrusions.

IQSEC1 functions with both ARF5 and ARF6 to control integrin endocytosis and focal adhesion disassembly[13,34]. Given IQSEC1 GEF activity-dependency for invasion (Supplementary Fig. 3d–g), we examined whether ARF5 and ARF6 are required for this process. While individual ARF depletion caused some changes that mimicked IQSEC1 loss, only combined ARF5/6 depletion reduced all aspects of 2D spindle shape and 3D growth and invasion (Fig. 3a–c, S4a). We examined whether individual or co-overexpression (OX) of ARF5 or ARF6, wild-type (WT) or fast-cycling mutants (Supplementary Fig. 4b, c)[13,35] induced invasive behaviours. Only ARF5/6 WT co-overexpression increased 2D spindle shape, 3D growth and 3D invasion (Fig. 3d–f, S4d–g). While fast-cycling mutants induced some 2D spindle behaviours, they failed to induce 3D effects, either alone or in combination, indicating that normal GTPase cycling was required. Crucially, IQSEC1 depletion abolished all effects of ARF5/6 co-overexpression (Fig. 3d–f, S4h). This identifies IQSEC1 as the major GEF-regulating growth and invasion with ARF5/6 as its major targets.

Despite identifying IQSEC1 GEF activity as essential for growth and invasion (Supplementary Fig. 3e–g) we did not observe global defects in GTP-loading of ARF5 or ARF6 upon IQSEC1 depletion (Supplementary Fig. 4i). As IQSEC1 v2 localised to a discrete pool at tips of protrusions (Fig. 2j), we reasoned that IQSEC1 may regulate a small but crucial ARF5/6 pool. We developed an imaging-based approach to determine the localisation of the IQSEC1–ARF complex. Endogenous IQSEC1 v2 localised with ARF5-mNeonGreen and ARF6-TagRFPT in tubulovesicular structures near the leading edge lamellipodium of spindle 2D cells (Fig. 3g, white arrowheads). In 3D, ARF5 displayed a prominent vesicular localisation in the acinus body, which overlapped with IQSEC1 in some regions, while ARF6 was prominent at cell–cell contacts lacking IQSEC1 (Fig. 3h, yellow arrowheads and white arrows, respectively). Both ARF5 and ARF6 could be found with IQSEC1 v2 in protrusion tips (Fig. 3h, white arrowheads).

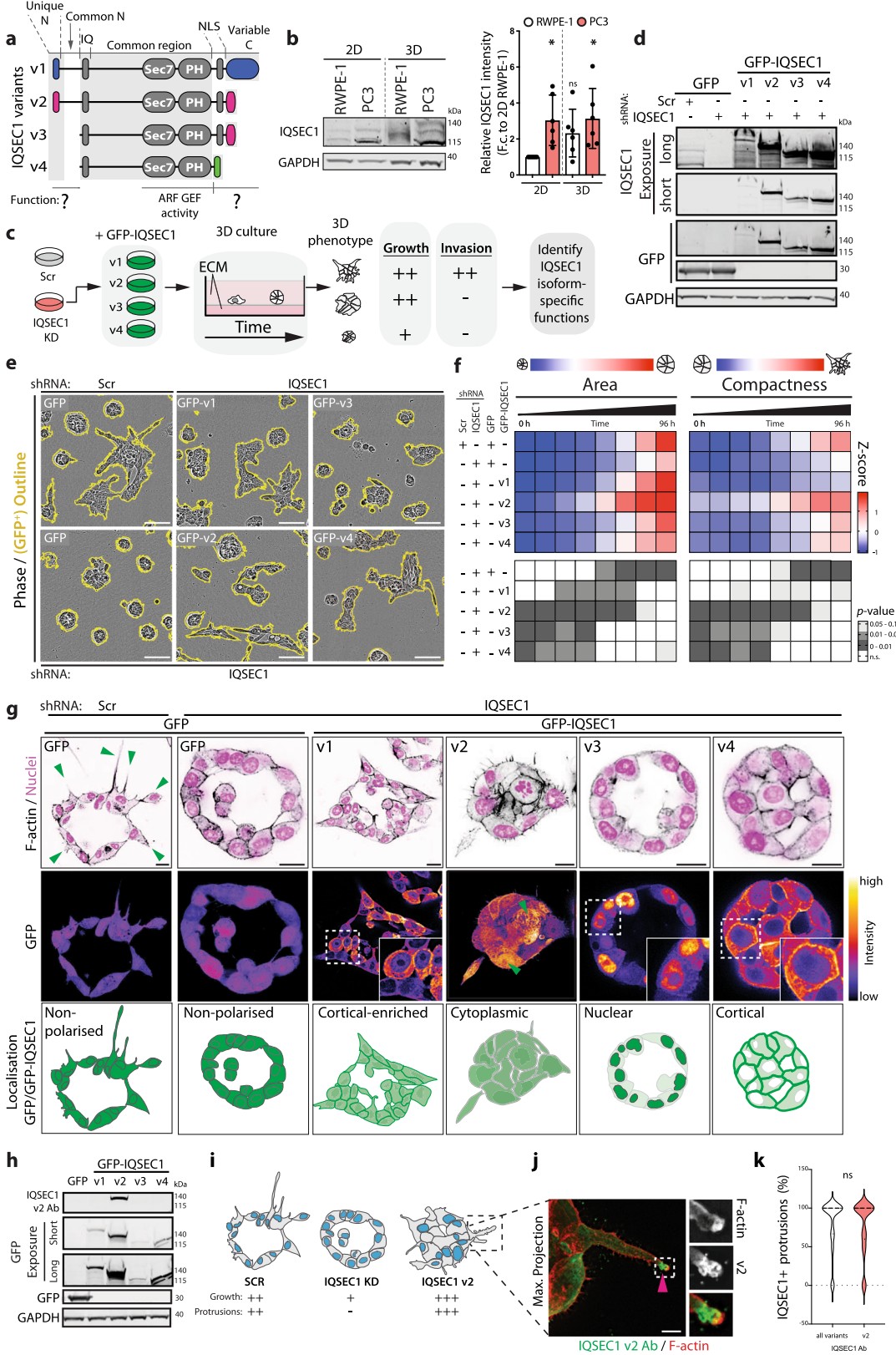

To identify the location of active ARF we expressed a sensor that detects GTP-loaded ARF proteins (GGA1 NGAT domain[36]) (Fig. 3i). GGA1 NGAT-TagRFP-T colocalized with ARF6 most notably in the tips of protrusions (Fig. 3j, green arrowheads), and with ARF5 in puncta in the body of the protrusion (white arrowheads) and the acinus body itself (white arrows).

To confirm that IQSEC1 was controlling this localised GTP loading of ARF5/6 we developed an automated method to analyse the percentage of overlap of the GGA1-NGAT probe with ARF5- or ARF6-positive puncta. We validated the sensitivity of this approach by first examining whether chemical inactivation of ARF GEF or GAP activity could be detected (Fig. 3i). We

**Fig. 2 IQSEC1 isoforms differentially regulate collective invasion. a** Schema, domain structure of IQSEC1 variants (v). Common domains, grey. Unique domains, colour: blue, v1; pink, v2 and v3; green, v4. **b** Western blot in 2D and 3D using anti-IQSEC1 or GAPDH antibodies. Relative expression of all IQSEC1 bands normalised to 2D RWPE1 is shown. Mean ± s.d., $n = 6$ independent experiments. p-values (Student's 2-tailed t-test), *$p ≤ 0.05$; n.s., not significant. **c** Schema, pipeline for identifying isoform-specific IQSEC1 functions. **d** Western blot of PC3 cells expressing GFP or GFP-IQSEC1 v1–v4, and either Scr or *IQSEC1* KD4 shRNA using anti-IQSEC1, GFP and GAPDH antibodies (all used on same membrane). Different exposures demonstrate expression of all variants. Upper and lower parts of same GFP blot demonstrate expression of GFP-IQSEC1 variants and GFP control. $N = 3$ independent experiments. **e** and **f** Phase images of acini (GFP-positive, yellow) from cells described in (**d**). Scale bars, 100 μm. Heatmap, area and compactness measurements Z-score-normalised to control values. p-values, one-way ANOVA, greyscale values as indicated. $n = 3$ independent experiments, 3 replicates/condition, 300–650 acini/condition in total. **g** PC3 acini described in (**d**) were stained for F-actin (black) and nuclei (magenta). Localisation of GFP-IQSEC1 can be appreciated from FIRE pseudo-coloured Look Up Table (FIRE LUT) (green arrowheads, cytoplasmic localisation). Magnified images are inset. $n = 3$ independent experiments. Scale bars, 20 μm. Cartoon, localisation of GFP-IQSEC1 variants in PC3 acini. **h** Western blot of PC3 cells expressing GFP or GFP-IQSEC1 v1–v4 using anti-IQSEC1 (v2-specific) or GAPDH (loading control) antibodies. Different exposures allow detection of all variants. Upper and lower parts of same GFP blot are shown to demonstrate expression of GFP-IQSEC1 variants and GFP control. **i** Schema, summary of the effect of *IQSEC1* KD and GFP-IQSEC1 v2 expression on growth and protrusive ability of PC3 acini. **j** Endogenous IQSEC1 v2 (green) co-stained with F-actin (red). Magnified images are shown. Arrowhead indicates localisation at protrusion tip. $n = 3$ independent experiments. Scale bars, 10 μm. **k** IQSEC1-positive protrusion tips were quantified (% positive/acinus) using antibodies which detect all variants (49 acini) or are specific for v2 (55 acini). Violin plot shows data distribution, $n = 2$ independent experiments. p-values, Student's 2-tailed t-test. n.s. not significant.

identified that a reported ARF6 inhibitor (NAV-2729)[29], but not the Cytohesin family ARF GEF inhibitor (Secin-H3)[37,38], also reduces IQSEC1-catalysed nucleotide exchange on ARF5 (Supplementary Fig. 4j, k). In cells, NAV-2729 significantly decreased GGA1-NGAT recruitment to both ARF5 and ARF6-positive puncta (Fig. 3k, l). In contrast, the ARF GAP inhibitor QS11 significantly increased GGA1-NGAT recruitment to ARF6, while decreasing recruitment to ARF5. Thus, our approach sensitively detects GTP-loading of specific ARF subpopulations (Fig. 3j–l). Applying this approach in IQSEC1-depleted cells we observed a robust decrease, but not complete loss, of recruitment of GGA1-NGAT to both ARF5 and ARF6 (Fig. 3j–l). These data confirmed that IQSEC1 controls GTP loading of a pool of ARF5/6 that is crucial for invasive activity (Fig. 3m).

We determined the morphogenetic effect of modulating ARF activity. Secin-H3 treatment, which minimally acts on IQSEC1-mediated ARF activation (Supplementary Fig. 4k) modestly decreased 3D growth, without affecting invasion (Supplementary Fig. 4l–o). ARFGAP inhibition (QS11)[27,38] increased 3D invasion, without affecting growth. However, in line with being a dual IQSEC1-ARF5/6 inhibitor, NAV-2729 treatment abolished 3D growth and invasion. Thus, ARF5 and ARF6 are the major targets of IQSEC1 GEF activity. Chemical or genetic inhibition of IQSEC1-directed ARF activation strongly attenuates 3D growth and invasion.

**IQSEC1 is a scaffold for Met and Akt signalling.** All IQSEC1 variants possess the SEC7-PH domains required to activate ARF GTP loading (Fig. 4a). The enhanced invasion-promoting capabilities of IQSEC1 v2 are conferred by the shared N-terminal extension (encoded by Exon 3; Fig. 4a). We used mass spectrometry protein–protein interaction analysis of IQSEC1 chimeras to identify domain-specific IQSEC1 binding partners. We classified interactors as unique to v1 or v2 N-termini (1N or 2N), shared between v1 or v2 N-terminal extension (Exon 3), to the core IQ-SEC7-PH region (Core), or specific to the v2 C-terminus (2C) (Fig. 4a). We prioritised interactors where multiple components of a complex could be identified. We ordered these by mRNA expression change from two pairs of PC3-derived sublines which possess epithelioid (PC3E, Epi) compared to mesenchymal properties (GS689.Li, EMT14)[39,40], to focus on invasion regulators (Fig. 4a–c). Using this approach, we identified the interaction of IQSEC1 with a number of oncogenic signalling pathways (mTORC2, RAF, NFkB), cytoskeletal and scaffolding complexes (Utrophin, 14-3-3), a phosphatase (PP6), and trafficking complexes (EARP, AP3/ARFGAP1). In addition to the

known interactors Calmodulin-1/-2, we also identified three transmembrane receptors (Met, LRP1 and SORL1), and other factors (TUBB6, FHOD1). We confirmed a number of these IQSEC1 interactors in independent immunoprecipitations, and interactor protein level alterations by western blotting of a series of PC3-derived sublines with varying invasive abilities (Fig. 4d, S5a–c).

We initially focused on transmembrane interactors of IQSEC1. IQSEC1 associated with Met, the receptor for HGF. This occurred through the core region, indicating that all IQSEC1 variants can potentially interact with Met (Fig. 4d, e, S5a). In contrast, two proteins interacted only with the invasion-inducing regions of IQSEC1: Low density lipoprotein receptor-Related Protein 1 (LRP1) and Sortilin-Related Receptor 1 (SORL1), members of the LDL Receptor family (Fig. 4d, e, S5a). LRP1 and SORL1 protein, however, displayed a mutually exclusive expression: LRP1 expression was elevated in invasive PC3 sublines (EMT14, GS689.Li), whereas SORL1 was associated with epithelial sublines (PC3E, Epi) and was anti-correlated with active pY1234/1235-Met (pMet) levels in cells (Supplementary Fig. 5b, c). In addition, LRP1 and Met both localised to the tips of invasive protrusions in 3D (Fig. 4f, g), and formed a complex irrespective of IQSEC1 expression levels (Supplementary Fig. 5d). These data suggest that LRP1 and Met form a complex in the tips of protrusions onto which IQSEC1 associates to promote invasion.

Supporting the enrichment of active Met and LRP1 in invasive PC3 sublines (Supplementary Fig. 5a–c), depletion of LRP1 or Met reduced spindle shape in 2D and attenuated 3D growth and invasion (Fig. 4h, i, S5e–g). LRP1 is a 600 kDa protein that is cleaved to release a 515 kDa extracellular fragment and an 85 kDa transmembrane receptor[41,42]. Expression of a GFP-tagged transmembrane LRP1 fragment (LRP1^TM-GFP) was not sufficient to induce invasion, suggesting both transmembrane and extracellular LRP1 fragments are required for invasion (Supplementary Fig. 5h). In contrast, SORL1 depletion increased spindle behaviours in 2D but did not significantly affect invasion or growth in 3D (Fig. 4h, i, S5g, i–k). These data support a positive role for LRP1/Met in invasion, while SORL1 may be antagonistic.

We examined downstream signalling from Met that drives invasion. We focused on signalling to the TORC2-Akt pathway, as multiple subunits of this complex (RICTOR, SIN1/MAPKAP1, PRKDC) associated with IQSEC1. RICTOR is an essential positive regulator of mTORC2[43,44], while SIN1 has been reported to inhibit mTORC2 activity[45]. Accordingly, RICTOR depletion robustly abolished Akt S473 phosphorylation (pAkt), spindle shape in 2D, and 3D invasion and growth (Fig. 4h, i, S5g, j–l).

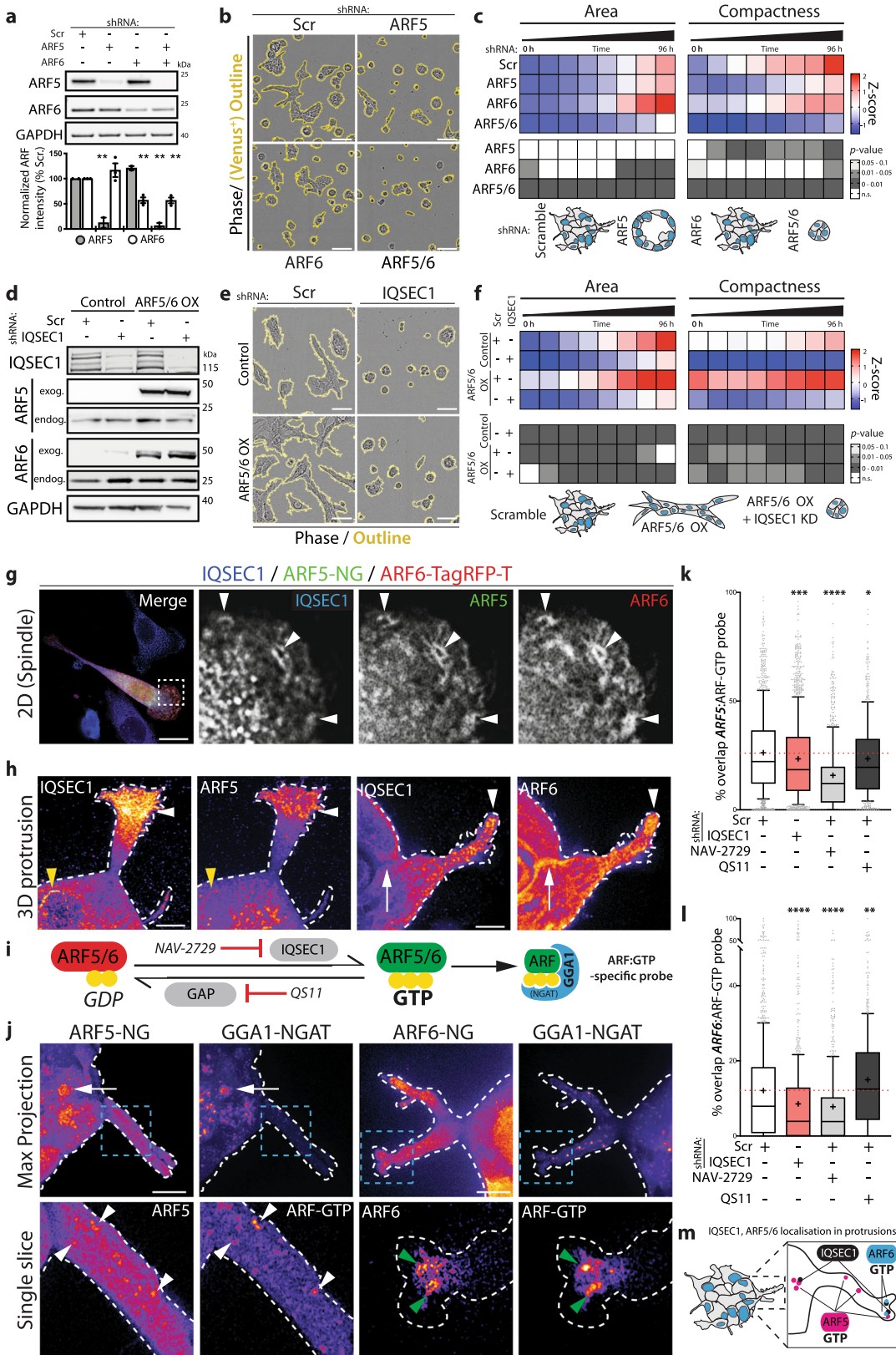

Notably, while SIN1 depletion decreased spindle behaviours in 2D, it robustly increased invasion in 3D, suggesting 3D-specific differences in this pathway (Fig. 4h, i, S5g, m). Finally, we examined ARFGAP1 as a potential antagonist of IQSEC1-ARF function. Unexpectedly, ARFGAP1 depletion phenocopied IQSEC1 depletion: decreasing pAkt levels, spindle shape in 2D,

modestly decreasing 3D growth, and profoundly inhibiting 3D invasion (Supplementary Fig. 5g, n–p). This suggests ARFGAP1 as an effector of ARF5/6 in regulating invasion and signalling to the mTORC2-Akt pathway (Supplementary Fig. 5q), similar to observation of dual effector-terminator functions of other ARFGAPs[46–48]. Thus, an IQSEC1-ARF5/6-ARFGAP1 complex

**Fig. 3 IQSEC1 activates ARF5/6 in distinct locations within protrusions. a** Western blot of cells expressing Scr, *ARF5* or *ARF6* shRNA using anti-ARF5, ARF6 and GAPDH (loading control) antibodies. ARF intensity normalised to Scr quantified. Mean ± s.d., $n = 3$ independent experiments. *p*-values, one-way ANOVA. **$p \leq 0.01$. **b** and **c** Phase images of acini (mVenus positive, yellow) described in (**a**). Scale bars, 100 μm. Heatmap, area and compactness measurements *Z*-score-normalised to control. *p*-values; one-way ANOVA, greyscale values as indicated. $n = 3$ independent experiments, 5 replicates/ condition, 2880–3188 acini/condition in total. Cartoon, acini phenotype representative of each condition. **d** Western blot of cells co-overexpressing (OX) mNeonGreen (mNG) and TagRFP-T (RFP) or ARF5-mNG and ARF6-RFP (ARF5/6), and Scr or *IQSEC1* KD4 shRNA. Anti-IQSEC1, ARF5, ARF6 and GAPDH (loading control for ARF5) antibodies used. $n = 2$ independent experiments. **e** and **f** Phase images of acini described in (**d**). Scale bars, 100 μm. Heatmap, area and compactness measurements *Z*-score-normalised to control. *p*-values, one-way ANOVA, greyscale values as indicated. $n = 2$ independent experiments, 4 replicates/condition, 1254–1567 acini/condition in total. **g** Cells co-overexpressing ARF5-mNG and ARF6-RFP (ARF5/6 OX) were stained for IQSEC1 v2. Merged and magnified image of spindle cell shown. Arrowheads, colocalization. $n = 3$ independent experiments. Scale bars, 20 μm. **h** Acini expressing ARF5-mNG or ARF6-mNG were stained for IQSEC1 (FIRE LUT). Yellow and white arrowheads, colocalization in juxtanuclear region and protrusive tips, respectively. Arrows, lack of colocalization. $n = 3$ independent acini imaged. Scale bars, 5 μm. **i** Schema, GTPase cycle, site of action of NAV-2729, QS11 and GTP-loaded ARF detection by GGA1-NGAT. **j** Acini expressing ARF5-mNG or ARF6-mNG and GGA1-NGAT-RFP were fixed and FIRE LUT of maximum projections and a single Z-slice is shown. Arrowheads, colocalization in protrusions, arrows, colocalization in cell body. $n = 2$–5 independent acini imaged. Scale bars, 10 μm. **k** and **l** Cells expressing GGA1-NGAT and **k** ARF5-NG or **l** ARF6-NG were transfected with Scr or *IQSEC1* KD4 shRNA then treated with NAV-2729 or QS11. % overlap of ARF and ARF-GTP probe/cell is shown in box-and-whiskers plot: 10–90 percentile; + mean; dots, outliers; midline, median; boundaries, quartiles. $n = 2$ independent experiments, 3 replicates/condition with **k** 541–1287 and **l** 390–798 cells quantified/condition. *p*-values, one-way ANOVA. *$p \leq 0.05$, **$p \leq 0.01$, ***$p \leq 0.001$ and ****$p \leq 0.0001$. **m** Schema, localisation of IQSEC1 and active ARFs in protrusions.

---

may control LRP1-Met signalling to mTORC2 during formation of invasive 3D protrusions.

**An IQSEC1–LRP1 complex modestly regulates Met endocytic trafficking.** Oncogenic Met signalling requires internalisation[9]. We investigated whether IQSEC1 controls HGF-Met signalling to Akt by controlling Met endocytic trafficking. HGF stimulation increased 3D growth and robustly induced invasion, which was substantially blunted by IQSEC1 depletion (Supplementary Fig. 6a–d). ARF5/6 co-overexpression resulted in a modest but consistent increase in pAkt levels, which was nonetheless attenuated to levels similar to parental cells by IQSEC1 depletion (Figs. 3d, 5a, b, S6a, b). This suggests that IQSEC1-ARF5/6 control signalling to the Akt pathway.

Paradoxically, IQSEC1-depleted cells displayed decreased total Met levels (Supplementary Fig. 6a, b), but an increased half-life of Met (Supplementary Fig. 6e), suggesting altered trafficking routes. We examined whether this was due to altered endocytosis and/or recycling of Met (Fig. 5c). We developed an image-based analysis of Met trafficking using a fluorescently conjugated anti-Met antibody (Fig. 5c, d). This allowed dual quantitation of internalisation levels and localisation to cellular sub-regions (membrane, cytoplasm, juxtanuclear). We detected some impact of IQSEC1 depletion on Met trafficking, though in all instances the magnitude of effect was modest. In control cells Met was efficiently labelled at the cell surface, appeared in peripheral endosomes after 10 min internalisation, then clustered in the juxtanuclear region by 30 min (Fig. 5d–f). IQSEC1-depleted cells showed a modest but significant delay in the internalisation and transit of Met from the periphery to the juxtanuclear region (Fig. 5d–f), representing a decrease in total Met internalisation levels. We observed no significant difference in recycling of internalised Met (Fig. 5e, 10 min), but defects in internalisation were re-apparent in extended time points of recycling that allowed for re-internalisation (30 min). Thus, IQSEC1 is required for efficient internalisation, but not recycling, of Met (Fig. 5f).

We examined whether the positive (LRP1) and negative (SORL1) interactors of IQSEC1-regulated invasion we identified may also function by controlling activated Met trafficking. LRP1/ SORL1 are regulators of endocytic sorting of a number of transmembrane proteins[49,50]. In control cells, puncta containing activated Met (pY1234/5-Met, pMet) were distributed throughout the cytoplasm. Treatment with HGF induced a reduction in peripheral pMet puncta, concomitant with puncta clustering in the juxtanuclear region (Fig. 5g, h), which was abolished by

IQSEC1 depletion (Fig. 5h–j). Consistent with a positive role for LRP1 in invasion LRP1, but not SORL1, depletion also blocked redistribution of pMet away from the cell periphery in response to HGF (Fig. 5h–j). These data reveal that IQSEC1 and LRP1 control Met-induced invasion by regulating Met transport to the juxtanuclear region (Fig. 5k), an essential function for signalling from Met[9]. The magnitude of effects of IQSEC1 depletion on trafficking, however, were modest. We therefore turned to potential effects on IQSEC1 in regulating signalling from Met.

**IQSEC1-ARF signalling controls phosphoinositide generation to induce invasion.** We next examined whether the major effect of IQSEC1-ARF5/6 on Met signalling is to contribute to Akt activation. Generation of PI(3,4,5)P$_3$ (PIP$_3$) from PI(4,5)P$_2$ is an essential event in the Akt signalling cascade, allowing PIP$_3$-dependent recruitment of Akt to the cortex, PIP$_3$-dependent PDPK1 phosphorylation of Akt T308[51] and the PIP$_3$-dependent release of SIN1 from RICTOR to allow Akt S473 phosphorylation[52]. PIP$_3$ generation requires the sequential action of PI4-kinases (PI4K), PIP5-kinases (PIP5K), and PI3-kinases (PI3K) to convert PI to PIP$_3$. All ARF GTPases can recruit and activate PIP5Ks[53–55], the latter of which there are three isoforms, PIP5K1A–C (PIP5K1α/β/γ). ARF6 can activate PI3K signalling in melanoma[56] and function to recruit PIP5K1α to Met[57]. We examined whether IQSEC1-ARF controls Akt activation by regulating PI(4,5)P$_2$ production as a precursor to PIP$_3$ generation.

We investigated the relationship between cell shape and PIP level and localisation using biosensors and antibodies to each lipid. We examined the ratio of lipid biosensor at the cell periphery, which includes a region of ~4 μm at or very close to the cell surface, to total expression. Accordingly, this analysis will only detect bulk general PIP ratios in the peripheral region, and not potential complex microtopologies of the cell membrane, which may contain subdomains with different PIP compositions. This analysis revealed that bulk general peripheral PI(4,5)P$_2$ levels were not different between cell shape classes (spindle, spread, round) (Supplementary Fig. 7a, b). In contrast, bulk peripheral PIP$_3$ was elevated in both spindle and spread shapes, but with alternate distributions. Spread cells had the highest bulk peripheral PIP$_3$ levels, which was uniformly peripheral. In contrast, though spindle cells displayed lower bulk peripheral PIP$_3$ than spread cells, bulk peripheral PIP$_3$ could be particularly noted in regions of protrusions in 2D and 3D (Supplementary Fig. 7a–f). This suggests that, at least at level of bulk analysis, a

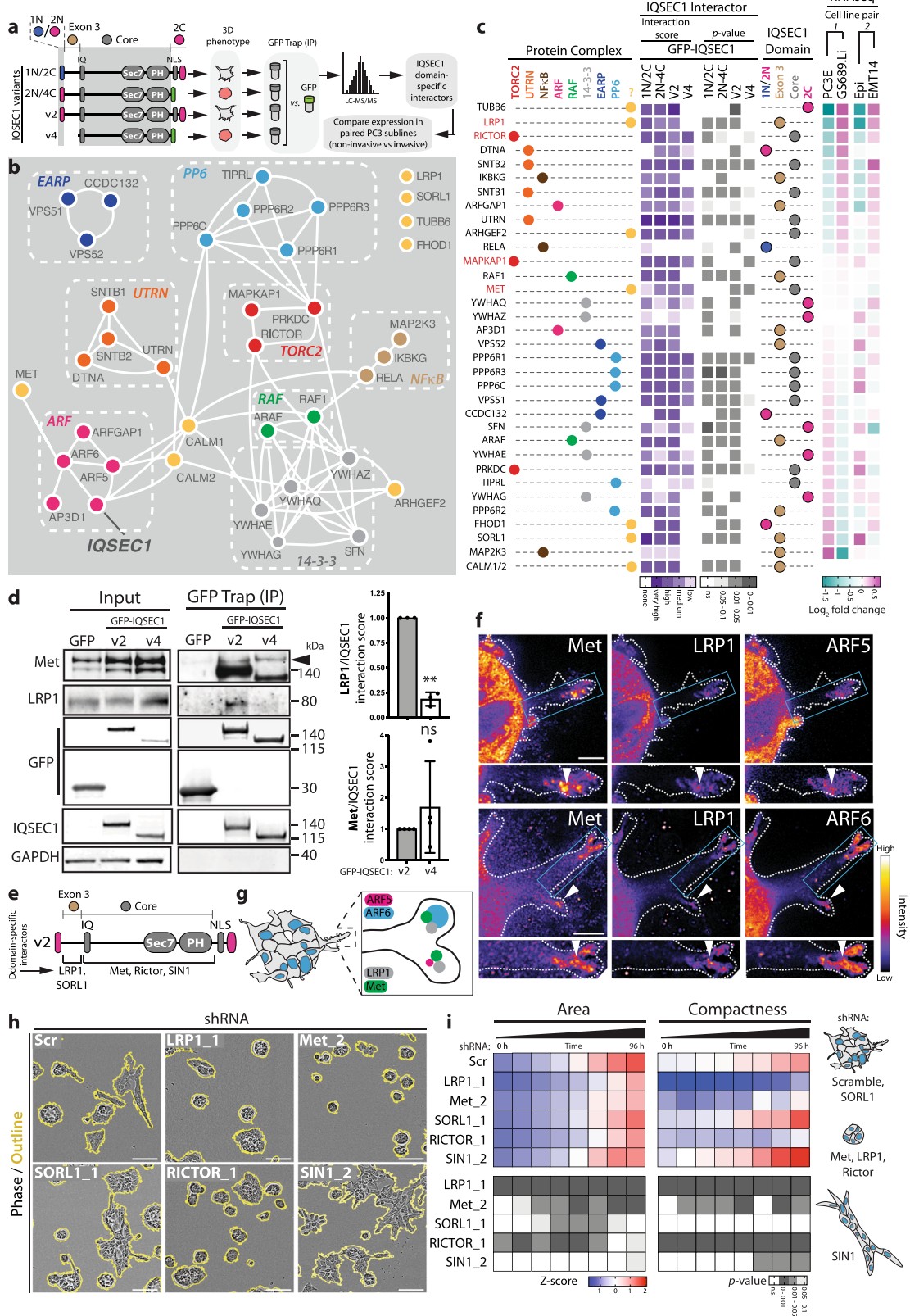

combination of level and asymmetry of PIP$_3$ promotes invasive function (Supplementary Fig. 7d).

To align total levels of PIPs with analysis of bulk peripheral distribution we validated that anti-PIP antibodies detected total cellular PIP levels, observing they are responsive to PI3K inhibition (LY294002) or activation (using myristoylated PI3K)

(Supplementary Fig. 7g–i). In contrast to PI3K inhibition, which decreases total and bulk peripheral levels of PIPs (Supplementary Fig. 7h, j), IQSEC1-depleted cells displayed an overall increase in total PI(4,5)P$_2$ and PIP$_3$ levels (Supplementary Fig. 7k), concomitant with a decrease in bulk peripheral PIPs (Figs. 6a, S7c) and the appearance of PIPs in intracellular compartments

**Fig. 4 IQSEC1 is a scaffold for Met signalling. a** Schematic, IQSEC1 chimeras and variants display distinct phenotypes in 3D culture. GFP-trap immunoprecipitation was performed on PC3 cells expressing these proteins followed by tryptic digestion "on-beads" and LC/MS/MS analysis. IQSEC1 domain-specific interactions were identified and sorted by mRNA expression compared in paired PC3 subclones. **b** STRING network analysis of IQSEC1-binding partners identified by MS visualised using Cytoscape. Most IQSEC1-binding partners could be clustered into 8 protein complexes (colour coded). **c** Schema, indicates the protein complex each binding partner is associated with. Heatmap shows the fold change of IQSEC1 interactors from panel **b** binding to different IQSEC1 domains over GFP control. Values are $-\log_2$(FC), very high = $-7$ to $-5$, high = $-5$ to $-2.5$, medium $-2.5$ to $-1$, low = $-1$ to $-0.05$, no binding >0.05. $p$-values; one-way ANOVA, greyscale values as indicated. The specific IQSEC1 domain to which each interactor binds is also depicted. Interactions were sorted according to the fold change in mRNA of non-invasive PC3 subclones (GS689.Li, EMT) compared to invasive subclones (PC3E, Epi) (RNAseq). **d** GFP-trap immunoprecipitation was performed on cells expressing GFP, GFP-IQSEC1 v2 or v4. Western blot analysis was carried out using anti-Met, LRP1, GFP, IQSEC1 and GAPDH (loading control for the LRP1) antibodies. Quantitation of LRP1/IQSEC1 and Met/IQSEC1 interactions for each GFP-trap are shown. Mean ± s.d., $n = 3$ and $n = 4$ independent experiments, respectively. $p$ values, one-way ANOVA. $^{**}p \le 0.01$ and n.s. not significant. **e** Cartoon, depicts domain-specific binding of IQSEC1 interactors. **f** PC3 acini expressing ARF5-mNG or ARF6-mNG were stained for Met and LRP1. FIRE LUT images are displayed with magnified images shown in lower panels. White arrowheads indicate localisation in protrusion tips. $n = 4$ independent acini imaged. Scale bars, 5 μm (upper) and 10 μm (lower). **g** Schema, colocalization of IQSEC1-ARF interactors in protrusive tips. **h** and **i** Phase images of PC3 acini expressing Scr, *LRP1_1, Met_2, SORL1_1, RICTOR_1* or *SIN1_2* shRNA. Scale bars, 100 μm. Heatmap, area and compactness measurements $Z$-score-normalised to control. $p$-values, one-way ANOVA, greyscale values as indicated. $n = 2$ independent experiments, 4 replicates/condition, 700–2400 acini/condition in total. Cartoon, depicts acinus phenotype representative of each condition.

---

(Supplementary Fig. 7a). This reduction of bulk peripheral $PIP_3$, in the absence of IQSEC1 was observed equivalently whether the cell cortex was defined using either F-actin or a cytoplasmic stain (CellMask) (Supplementary Fig. 7l), and irrespective of cell shape (Supplementary Fig. 7l), representing that bulk differences in PIP levels were not simply a consequence of altered shape. Reduced bulk peripheral PIP3 levels in IQSEC1-depleted was comparable to the magnitude of PI3K-inhibited cells (Supplementary Fig. 7j). These data suggest that IQSEC1-ARF-PIP5K is not essential for general $PI(4,5)P_2$-$PIP_3$ generation, but rather that a LRP1–Met–IQSEC1 v2 complex promotes bulk peripheral $PIP_3$ production that leads to invasive protrusion formation (S7d, g).

IQSEC1 v2 promotes single cell spindle shape (Supplementary Fig. 2d) and 3D invasion (Fig. 2e, f), while IQSEC1 v4 promotes round cell shape. No difference in bulk peripheral $PI(4,5)P_2$ levels between cell shape classes (spindle, spread, round) was observed (Supplementary Fig. 7a, b). Accordingly, rescue of IQSEC1 depletion with either IQSEC1 v2 or v4 partially restored total $PI(4,5)P_2$ levels to those observed in control cells (Supplementary Fig. 7m). In contrast, the invasion-inducing IQSEC1 v2 supported more robust activation of $PIP_3$ generation than could IQSEC1 v4 (Supplementary Fig. 7m). Given that IQSEC1 v2 induces a switch to spindle shape, this suggests that IQSEC1 v2-directed bulk peripheral production of $PIP_3$ drives invasion.

We next examined the components of $PIP_3$ generation that drive invasion. Isotype-selective Class I PI3K inhibitors revealed that either PI3Kβ or PI3Kδ inhibition decreased pAkt levels, while PI3Kα or PI3Kγ inhibition paradoxically increased pAkt levels (Supplementary Fig. 8a). However, only PI3Kβ or Akt inhibition significantly attenuated 3D growth and invasion (Supplementary Fig. 8b–d). Depletion of PIP5K1A-C revealed that PIP5K1β depletion showed the most robust decrease in growth, invasion, and pAkt levels (Supplementary Fig. 8e–g). These data suggest a PIP5K1β → PI3Kβ → Akt pathway for 3D growth and invasion.

Our data suggest that IQSEC1-ARF5/6 controls the PIP production required for PIP5K1β→ PI3Kβ→ Akt signalling during invasion (Supplementary Figure 7e). We tested whether generalised cortical targeting of this PIP-Akt pathway could overcome the need for IQSEC1 (Fig. 6b). We targeted each of PIP5K1β, PI3Kβ, Akt1 to the cortex through addition of a myristoylation sequence, and inhibited IQSEC1 by shRNA or chemical means (NAV-2729) (Fig. 6b, c).

Myristoylation of PI3Kβ (PI3Kβ$^{myr}$) or PIP5K1β (PIP5K1β-$^{myr}$), but not PIP5K1α (PIP5K1α$^{myr}$), increased Akt phosphorylation, 3D growth and invasion, with the most robust increase

occurring upon PI3Kβ$^{myr}$ expression (Fig. 6c–f). As expected in control cells, IQSEC1 depletion reduced pAkt levels and attenuated 3D growth and invasion (Fig. 6c–f). Myristoylation of PI3Kβ (PI3Kβ$^{myr}$) or PIP5K1β (PIP5K1β$^{myr}$), but not PIP5K1α (PIP5K1α$^{myr}$), increased 3D growth and invasion with the most robust increase occurring upon PI3Kβ$^{myr}$ expression (Fig. 6c, e, f). Strikingly, IQSEC1 perturbation (shRNA, NAV-2729) had a disproportionate effect on invasion versus growth. While growth was blunted, in IQSEC1-perturbed conditions invasion was abolished (Fig. 6c, e, f). NAV-2729 treatment blocked the ability of all cells in 3D to become acini. Single cells initially increased slightly in area and adopted irregular shapes, manifested in increased compactness, but not in invasion. Thus, generalised cortical recruitment of PIPK signalling is sufficient to drive growth, but not invasion. Invasion requires IQSEC1-dependent ARF activity at invasive protrusions (Fig. 6i).

**IQSEC1-ARF regulates localisation and activation of Akt**. Akt signalling showed a different influence on 3D behaviours than that observed for PI3Kβ or PIP5K1β. Our data suggest that IQSEC1 may contribute to Akt S473 levels by at least two mechanisms: influencing PIP5K1β-directed $PI(4,5)P_2$ production, and interaction with RICTOR-SIN1 (Supplementary Fig. 7e). While IQSEC1 depletion lowered endogenous pAkt levels, expression of myristoylated Akt1 (Akt1$^{myr}$) overcame the requirement of IQSEC1 for pS473 phosphorylation (Fig. 6d, g, h). As predicted, general cortical targeting of Akt1$^{myr}$ robustly increased 3D area compared to control cells, but not invasion (Fig. 6g, h). Strikingly, although Akt1$^{myr}$ no longer required IQSEC1 for phosphorylation events normally indicating 'activeness' (i.e. pS473-Akt, Fig. 6d), IQSEC1 depletion still reversed the Akt1$^{myr}$ -induced 3D growth increase to levels below control cells, and abolished invasion; NAV-2729 was similarly an inhibitor of these processes (Fig. 6g, h). Thus an effector of $PIP_3$ signalling, Akt, requires IQSEC1 for full signalling output.

As experimentally 'active' Akt (asymmetrically cortically targeted Akt1$^{myr}$; as defined by pS473 levels) was unable to induce oncogenic signalling in IQSEC1-depleted cells we examined whether this was due to altered Akt localisation. In 2D, pAkt localised to cortical regions and to puncta distributed through the cytoplasm (Supplementary Fig. 8h). In 3D, a pool of pAkt was enriched in protrusion tips (Supplementary Fig. 7f, white arrowheads), as well as to puncta throughout the protrusion and acinus body. In contrast, IQSEC1-depleted cells displayed an increased cortical pAkt aggregate size, but with strongly reduced intensity (Supplementary Fig. 8h–k). These data

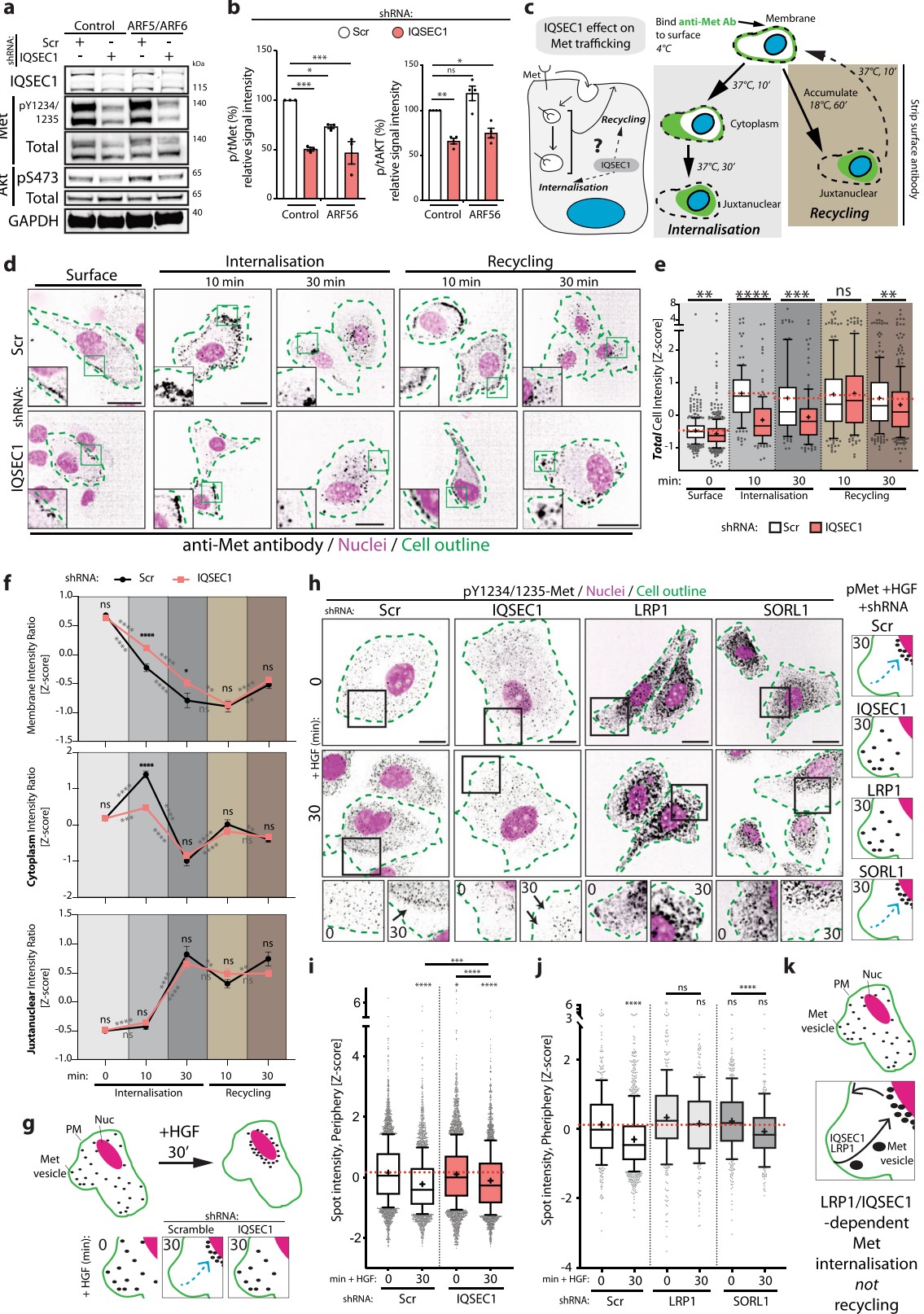

reveal that IQSEC1 is required for the signalling from, and asymmetry of, the PIP$_3$ effector Akt.

**IQSEC1 regulates growth and invasion in vitro and in vivo.** Our data indicate that LRP1-Met-IQSEC1-promoted enrichment of ARF5/6-dependent PIP$_3$ signalling to induce invasion rather than growth. We tested the generalisability of IQSEC1 inhibition to inhibit growth and invasion mechanisms in commonly used prostate cancer models. mRNA expression and western blotting indicated that, with the exception of LRP1 in 22Rv1, all components of the LRP1-Met-IQSEC1-ARF5/6 pathway are expressed in examined prostate cancer cell lines (Supplementary Fig. 9a, b).

**Fig. 5 IQSEC1–LRP1 complex regulates Met endocytic trafficking. a** Western blot of cells co-expressing mNG and RFP or ARF5-mNG (Control) and ARF6-RFP (ARF5/6) with Scr or *IQSEC1* KD4 shRNA. Anti-IQSEC1, phospho-Y1234/1235 Met, Met, phospho-S473 Akt, Akt, and GAPDH (sample control) antibodies used. **b** Quantitation of phospho/total Met and phospho/total Akt expression is presented as signal intensity relative to control. Mean ± s.d., $n = 3$ independent experiments. *p* values; one-way ANOVA. **c** Schema, effect of IQSEC1 on Met trafficking. **d** Cells expressing Scr or *IQSEC1* KD4 shRNA were incubated with a Met-647 antibody (4 °C) (Surface) then stimulated with HGF (Internalisation). Chloroquine was added to allow accumulation of surface-derived Met (black) (1 h, 17 °C) prior to HGF stimulation (Recycling). Cells were stained with F-actin (green, outlines) and Hoechst (nuclei, magenta). Magnified images are inset. Scale bars 20 μm. **e** Intensity of Met/cell was quantified (*Z*-score normalised); shown in box-and-whiskers plot: 10–90 percentile; + mean; dots outliers; midline median; boundaries, quartiles. $n = 2$ independent experiments, 4 replicates/condition, 421, 546, 122, 126, 93, 120, 150, 136, 219 and 282 cells quantified in total. *p* values: Welsh's 2-tailed *t*-test. **f** Line graphs show relative regional intensities compared to whole cell intensity of Met staining (*Z*-score normalised). Mean ± s.e.m., $n = 2$ independent experiments, 4 replicates/condition, 421, 122, 93, 150 and 219 (Scr) and 546, 126, 120, 136, and 282 (*IQSEC1* KD4) cells quantified in total. *p* values: Welsh's 2-tailed *t*-test. **g** Schema, sub-cellular re-localisation of active Met upon HGF treatment. **h** Cells expressing Scr, *IQSEC1 KD4, LRP1* or *SORL1* shRNA were stimulated with HGF then stained for phospho-Met (black), F-actin (green, outlines) and Hoechst (nuclei, magenta). Magnified images are shown. Arrows, reduction of pMet in periphery. Scale bars 20 μm. Cartoon, sub-cellular localisation of active Met. **i** and **j** Quantitation of spot intensity (*Z*-score normalised) is shown for images in (**h**) in box-and-whiskers plot: 10–90 percentile; + mean; dots outliers; midline median; boundaries quartiles. **i** $n = 7$ independent experiments, 3 replicates/condition, 3519–5185 cells/condition and **j** $n = 2$ independent experiments, 4 replicates/condition, 239–988 cells/condition. *p*-values; one-way ANOVA. **k** Schema, regulation of Met internalisation, but not recycling, by IQSEC1 and LRP1. All *p* values; n.s. not significant, *$p \leq 0.05$, **$p \leq 0.01$, ***$p \leq 0.001$ and ****$p \leq 0.0001$.

Chemical inhibition (NAV-2729, Supplementary Fig. 9c–h) or genetic depletion (Supplementary Fig. 10a–f) of IQSEC1 attenuated growth and/or invasion in a range of 3D cancer cell models. This included upon ectopic OX of the LRP1–IQSEC complex in cells with low endogenous levels (LRP1$^{TM}$-GFP in 22Rv1, GFP-IQSEC1 v2 in DU145), HGF treatment of the mixed morphology DU145 cultures to resemble the spindle-type invasion of PC3 (Supplementary Fig. 9, S10), highly invasive human breast cancer cells (MDA-MB-231), murine pancreatic ductal adenocarcinoma cells (PDAC; KC-PTEN, K-rasG12D/PTEN-null[58]) and patient-derived PDAC cells (TKCC-07) (Supplementary Fig. 10a, c–f). Thus, IQSEC1 is required for growth and invasion across a number of 3D cell models from different cancer types.

We examined the in vivo role of IQSEC1 by intraprostatic xenograft of IQSEC1-depleted PC3 cells (Fig. 7a). While IQSEC1 depletion did not significantly attenuate tumour incidence, tumour area and volume were significantly reduced (Fig. 7b–e). As predicted from our 3D in vitro studies, metastatic activity was strongly decreased in IQSEC1-depleted cells. Both the incidence and number of macrometastases were significantly decreased in IQSEC1-depleted cells (Fig. 7f, g). Wide-spread dissemination of macrometastases was observed in controls cells. In the few mice presenting macrometastases in IQSEC1-depleted cells, these were limited to prostate proximal lymph nodes (with the exception of a singular diaphragm-located tumour) (Fig. 7h), and showed no difference in proliferation or apoptosis to controls, confirming a bona fide effect on movement (Fig. 7i, j). These data indicate an essential requirement for IQSEC1 in metastasis in vivo.

Elevation in IQSEC1, ARF5 and ARF6 levels was associated with clinical outcome in prostate cancer across 12 studies, representing 2910 patients (Fig. 8a). Increased Copy Number (CN) of IQSEC1 occurred most frequently in advanced prostate cancers (Fig. 8b, CR, castrate-resistant; NE, neuroendocrine), associated with a gain in 3p Status (Fig. 8c; IQSEC1, 3p25.2-25.1). Primary prostate tumours with elevated IQSEC1 were of significantly higher grade (Fig. 8d), were associated with tumour-bearing lymph node positivity (Fig. 8e) and metastases (Fig. 8h). Mirroring the ablation of wide-spread metastasis of IQSEC1-depleted PC3 xenografts (Fig. 7h), IQSEC1 increase occurred most frequently in samples from diverse metastatic sites, but particularly bone, liver and lymph node (Fig. 8g). IQSEC1 increase occurred exclusively with androgen deprivation therapy, retained presence of tumour after therapy, and increased levels of

the androgen receptor V7 variant, a major mechanism for escape from androgen deprivation (Fig. 8h–j). A clinical indicator of disease, serum PSA levels, was elevated in patients displaying combined IQSEC1 and ARF5/6 elevation (Fig. 8k). This suggests a clinical association of IQSEC1–ARF5/6 gain with therapy resistance and metastasis.

We examined the association of IQSEC1, ARF5, and ARF6 elevation with frequent genomic alterations in prostate cancer. IQSEC1 increase was associated with amplifications particularly in AR and MYC and to a lesser extent with loss of PTEN, and prominently with TP53 mutation, but not common gene fusion events (Fig. 8l–n, S11a). ARF6 increase followed a similar profile, while ARF5 increase was associated with a broader range of mutational events. IQSEC1 allowed robust stratification of prostate cancer patients, stratifying a 24-month decrease in median overall survival in IQSEC1-elevated patients (Fig. 8o, 665 patients). Inclusion of ARF5/6 increase did not change the median overall survival of this group, but rather extended the median survival of the control arm, resulting in a 35-month survival difference compared to the IQSEC1-ARF5-ARF6 elevated group. IQSEC1 increase was similarly associated with TP53 mutation and MYC amplification across the pan-cancer The Cancer Genome Atlas (TCGA) dataset, representing 10,449 patients (Supplementary Fig. 11a–d). Strikingly, IQSEC1, ARF5 and ARF6 increase similarly stratified patient survival when all tumour types were considered together, providing an exceptional >9-year (111 months) median survival increase for non-IQSEC1-ARF5/6-amplified patients (across 9307 patients) (Fig. 8p).

Comparison of matched normal–tumour tissue from the TCGA prostate cohort ($n = 52$) showed a significant increase in overall IQSEC1 mRNA in tumours (Fig. 8q). This was accompanied by a switch from the non-invasive IQSEC1 v1 isoform, to the pro-invasive IQSEC1 v2 in tumours (Fig. 8r). Analysis of the pan-cancer TCGA dataset for tumour types with profiled normal tissue revealed that four tumour types (kidney, KICH; liver, LIHC; prostate; sarcoma, SARC) displayed IQSEC1 mRNA higher than normal tissue (Fig. 8s). However, seven tumour types showed that rather than an increase in overall IQSEC1, instead tumours underwent an isoform switch from v1-to-v2 (Fig. 8s). This demonstrates that isoform switching to IQSEC1 v2 is a major event associated with tumourigenesis in patients.

Finally, we examined the association of IQSEC1 mRNA levels across the pan-cancer TCGA dataset with common tumour-associated signalling pathways from reverse phase protein array

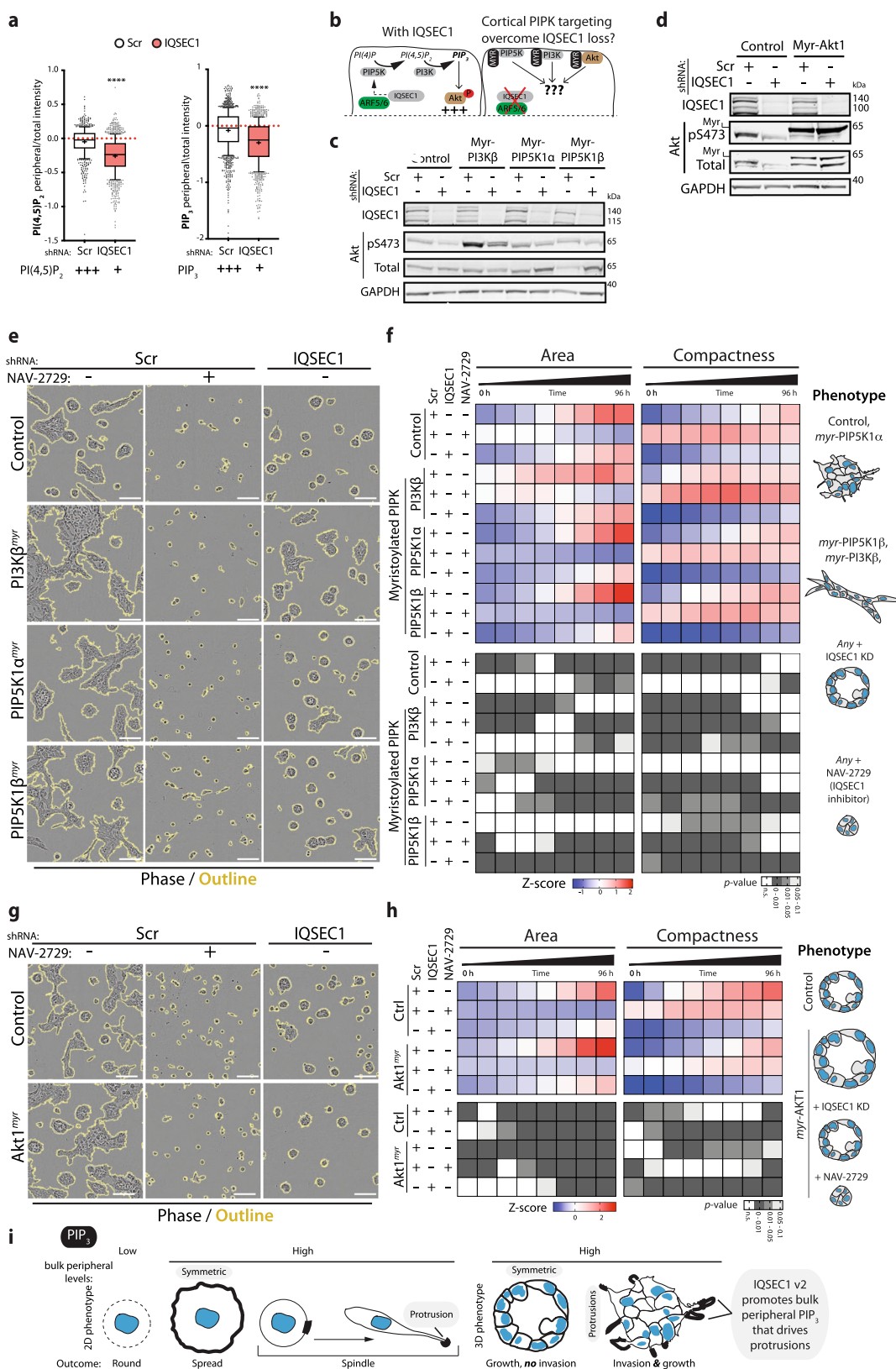

(RPPA) data. Within each cancer type IQSEC1 expression was divided into high and low using a median split of mRNA levels, and protein differences between each group that consistently and significantly trended in the same direction across a quarter of all cancer types was calculated (Supplementary Fig. 11e). Mirroring IQSEC1 regulation of phosphoinositide signalling in PC3 cells, a clear PI3K-AKT signature was associated with high IQSEC1 levels across ten tumour types (Fig. 8t). Together, these data indicate a role for tumour-associated isoform switching to IQSEC1 v2 to promote PI3K-AKT signalling and metastasis across cancer types, resulting in treatment resistance and a robust decrease in patient survival.

**Fig. 6 IQSEC1-ARF signalling controls phosphoinositide generation during invasion. a** Quantitation of $PI(4,5)P_2$ or $PIP_3$ in the presence or absence of IQSEC1 is shown in box-and-whiskers plot: 10–90 percentile; + mean; dots outliers; midline median; boundaries quartiles. Values, peripheral/total intensity/cell, $n = 3$ independent experiments, 4 replicates/condition, 1674/1959 (Scr and *IQSEC1* in upper panel) and 2684/2083 (Scr and *IQSEC1* in lower panel) cells quantified. $p$-values; one-way ANOVA. ****$p \le 0.0001$. **b** Cartoon, PIPK targeting in presence or absence of IQSEC1. **c** Western blot of PC3 cells expressing Myr-FLAG-Cre (Control), Myr-FLAG-PIP5Kα, Myr-FLAG-PIP5Kβ or Myr-FLAG-PI3Kβ and either Scr or IQSEC1 shRNA (KD4). Anti-IQSEC1, phospho-S473 Akt, total Akt and GAPDH (loading control for IQSEC1) antibodies were used. $n = 2$ independent experiments. **d** Western blot of PC3 cells expressing Myr-FLAG-Cre (Control) or Myr-Akt1 and either Scr or *IQSEC1* KD4 shRNA using anti-IQSEC1, phospho-S473 Akt, total Akt and GAPDH (loading control for IQSEC1) antibodies. $n = 2$ independent experiments. **e** and **f** Phase images of PC3 acini described in (**c**) are shown. Scr acini were also treated with NAV-2729 (IQSEC1 inhibitor). Scale bars, 100 μm. Heatmap, area and compactness measurements $Z$-score-normalised to control. $p$-values; one-way ANOVA, greyscale values as indicated. $n = 3$ independent experiments, 4 replicates/condition, 1693–2435 acini/condition in total. Cartoon, acini phenotype representative of each condition. **g--h** Phase images of PC3 acini described in (**d**) are shown at 96 h. Scr acini were treated with IQSEC1-inhibiting compound NAV-2729. Scale bars, 100 μm. Heatmap, area and compactness measurements $Z$-score-normalised to control. $p$-values; one-way ANOVA, greyscale values as indicated. $n = 2$ independent experiments, 4 replicates/condition, 1287–2363 acini/condition in total. Cartoon, acini phenotype representative of each condition. **i** Schema, summarises the relationship between location and level of peripheral $PIP_3$ and 2D and 3D PC3 phenotype.

## Discussion

Our results clarify a long-held conundrum of how the same signalling pathways lead to distinct biological outputs: alternate isoforms of the same RTK effector, IQSEC1, differentially localise to control ARF GTPase-dependent phosphoinositide metabolism at these distinct locales.

We describe that IQSEC1 is a key regulator of bulk peripheral $PIP_3$ signalling to promote invasion. Our data support a model whereby pro-invasive IQSEC1 isoforms form a complex with the HGF receptor Met and the endocytic receptor LRP1 at membranes that will develop into invasive protrusions (Fig. 7k). IQSEC1 activates GTP loading on ARF6 to (1) stimulate PIP5K-mediated $PI(4,5)P_2$ production, which is (2) a precursor to PI3K-mediated $PIP_3$ production to (3) promote invasive protrusion formation. Elevation of bulk $PIP_3$ levels also triggers cell growth, concomitant with internalisation of Met and (4) activation of pAkt downstream of the mTORC2 complex. Met and pAkt are retrograde transported in the body of protrusions on ARF5 endosomes to a juxtanuclear signalling compartment, essential for growth and invasion. Inhibiting this IQSEC1-mediated ARF5/6 activation abolishes the collective growth and invasion/metastasis in vitro and in vivo.

Multiple isoforms of IQSEC1 exist through combination of alternate start sites and alternate splicing, though the regulators of these events are unknown. Such N-terminal and C-terminal extensions operate in a hierarchy; N-terminal extensions provide association with endocytic co-receptors LRP1 and SORL1, while alternate C-termini can either positively or negatively enhance invasive activity. SORL1 and LRP1 are members of the LDLR family, co-receptors involved in endocytic control of a milieu of processes[41]. SORL1 and LRP1 shared the same N-termini binding region, were anti-correlated in their expression between invasive and non-invasive cells, and possessed antagonistic effects on invasion. This supports a notion that the function of IQSEC1 is highly contextual, dependent on not only which IQSEC1 transcript is expressed, but also the repertoire of potentially competitive interactors expressed. It is notable switching to the pro-invasive IQSEC1 v2 occurs in at least a quarter of all sampled tumour types, suggesting this may be a major mechanism underpinning tumourigenesis.

LRP1 is a multifunctional endocytic receptor involved in a number of biological functions[41]. We report this as a crucial pro-invasive partner of IQSEC1 and Met. Endogenous LRP1 and Met were found in a complex irrespective of IQSEC1, suggesting that IQSEC1 binds an existing complex to couple their endocytic itinerary. This likely also requires the function of the extracellular cleaved 515 kDa fragment of LRP1, as the transmembrane fragment alone is insufficient to drive invasion when expressed

ectopically. LRP1 may help generate $PIP_3$ during invasion through its ability to form an LRP1–Rab8a:GTP–PI3K complex, to control Akt signalling on endosomes, as it does in other contexts[59].

The role of phosphoinositides, particularly $PIP_3$, in regulating leading edge formation or protrusion activity has been extensively reported[60–67], including ARF6 being a driver of PI3K enrichment in melanoma cell protrusions[57]. Our data further supports central roles for ARF GTPases in invasion. Protrusive invasion requires a discrete pool of protrusive activity concomitant with global inactivity in surrounding regions, akin to pushing out from single point inside a balloon rather than simultaneously from all surfaces. IQSEC1 v2 localised with Met and LRP1 and promoted elevation in bulk $PIP_3$ production resulting in invasive protrusions. Such exquisite spatial enrichment may explain why the most prominent peripherally recruited IQSEC1 isoform (v4) fails to induce robust invasion, and perhaps why we see significant, but modest global GTP loading changes in ARF5/6 upon IQSEC1 depletion. The identity of invasion-specific $PIP_3$ effectors remains to be demonstrated, but one likely candidate is Rac1, which is dependent on $PIP_3$ for activation[68].

Why are both ARF5 and ARF6 required? Although ARF5 phenotypes most closely mimicked that of IQSEC1, there was still a dependency on ARF6. ARFs can act in pairs[69], and our data suggests that ARF5/6 may act at sequential steps: ARF6 at the cortex, and ARF5 in endosomes. It is interesting to note that previously described fast-cycling mutants of either ARF[35] were unable to induce collective morphogenesis like WT ARFs, suggesting that precise control over GTP cycling is essential for collective invasion. Indeed, a key effector identified was ARFGAP1, which in our hands functioned as an effector rather than an antagonist of ARF signalling, similar to the dual effector terminator function described for other ARFGAPs[46–48]. That this GAP associates with the GEF supports the notion of tight control over GTP cycling. Whether ARF5/6 both, or separately, interact with the two identified effectors (ARFGAP1, PIP5K1β), is unclear.

The ARF GTPase-dependent recycling of Met promotes sustained ERK signalling required for migration in other cells[57,70]. Internalisation of Met is key to full oncogenic output[8,9]. IQSEC1 has modest effects on internalisation and juxtanuclear trafficking of Met, suggesting the major role of IQSEC1 is to couple Met to its downstream component Akt. Met recycling was not perturbed in IQSEC1-depleted cells, suggesting another ARFGEF may promote this function. The $PI(4,5)P_2/PIP_3$-dependent Cytohesin-1 ARFGEF is one candidate for this, as we note upregulation in PC3, and it is required downstream of Met for invasive activity in other cells[71]. Cytohesin-1 alternate splicing controls its recruitment to membranes by controlling differential affinity for

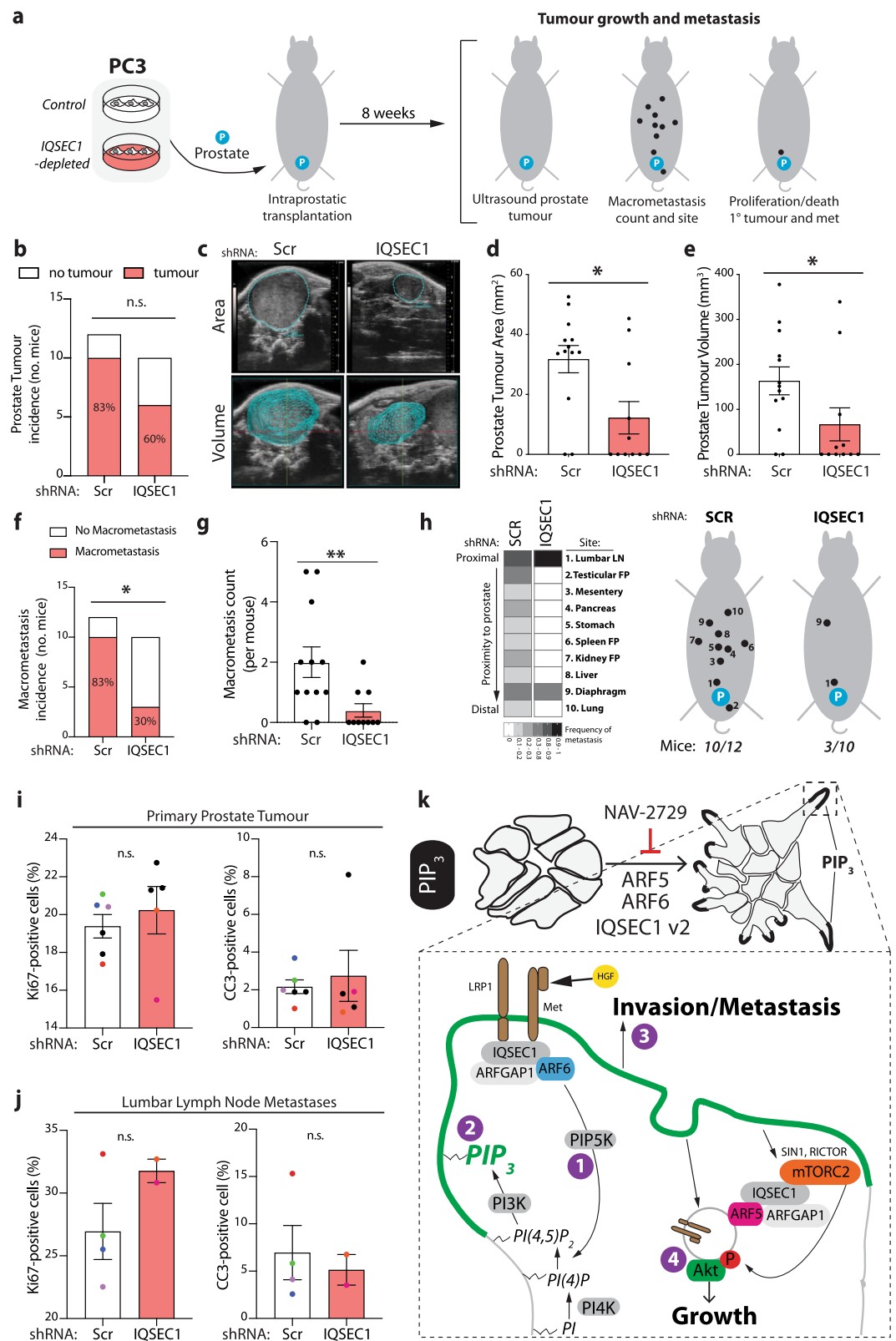

PI(4,5)P$_2$ or PIP$_3$[72,73]. Such a GEF would likely act subsequently to IQSEC1-directed control of peripheral PI(4,5)P$_2$ levels. Thus, IQSEC1 may sit at the top of a cascade of ARF GEFs controlling Met endocytosis, recycling, and signal output. This may explain why IQSEC1 inhibition alone can reverse ARF5/6 OX, despite co-expression of numerous ARFGEFs in PC3 cells.

A striking result is that although IQSEC1-ARF5/6 would classically be considered 'upstream' of PI(4,5)P$_2$, PIP$_3$, and Akt, inhibition of IQSEC1 nonetheless counteracts the experimental activation of this pathway at multiple levels. Our data suggests that this is due to alternate isoforms of IQSEC1 possessing different locations, thereby likely influencing where downstream

**Fig. 7 IQSEC1 regulates growth and invasion in vivo and IQSEC1-ARF signalling is associated with poor patient outcome. a** Schema, intraprostatic transplantation of PC3 cells expressing either Scr or *IQSEC1* KD4 shRNA into CD1-nude male mice. Tumour growth was determined after 8 weeks using ultrasound and the incidence and location of macromets counted. **b** Graph shows prostate tumour incidence (in total number of mice) in mice transplanted with PC3 cells expressing either Scr (control, 12 mice) or *IQSEC1* KD4 shRNA (10 mice). *p*-values (Chi-squared): n.s. not significant. **c** Ultrasound images show area and volume of representative prostate tumours in mice injected with PC3 cells expressing either Scr or *IQSEC1* KD4. **d** and **e** The **d** area and **e** volume of each prostate tumour detectable by ultrasound was measured. Values, mean ± s.d. *p*-values; Mann–Whitney test (2-tailed). *$p \leq 0.05$. Note that one mouse in each condition had two primary prostate tumours. Control, 12 mice, 13 tumours and *IQSEC1* KD4 shRNA, 10 mice, 11 tumours. **f** and **g** Macrometastasis incidence was counted and is presented as (**f**) number of mice and (**g**) count per mouse, values, mean ± s.e.m. 12 and 10 mice were transplanted with PC3 cells expressing Scr or *IQSEC1* KD4 shRNA, respectively. *p*-values (Chi-squared and Mann–Whitney test (2-tailed) respectively): *$p \leq 0.05$, **$p \leq 0.01$. **h** Heatmap shows the frequency at which metastasis occurred at locations with different proximity to the prostate. Cartoon, summarises the locations metastasis was observed. **i** and **j** The (**i**) tumour and (**j**) lymph node (LN) sections were stained with anti Ki67 and Cleaved Caspase 3 (CC3) antibodies and the percentage of positive cells quantified. Each mouse is represented by a differently coloured point on the bar graphs. Values, mean ± s.e.m. Sections were taken from 6 (Scr) and 5 (IQSEC1 KD4) tumours and from 4 (Scr) and 2 (IQSEC1 KD4) lymph nodes. *p*-values; Mann–Whitney test (2-tailed). n.s. not significant. **k** Schematic depicting the signalling mechanisms involved in IQSEC1 dependent model of prostate cancer growth and invasion.

ARF-dependent PIP production can occur. We note that our analysis approaches for PIP localisation at the periphery detects only bulk location of the PIP, and does not allow for detection of discrete subdomains containing different PIPs that may occur because of microtopologies of the cell surface. It remains to be demonstrated exactly how this IQSEC1-ARF module may control PIP metabolism spatially at the cell surface.

Critically, we demonstrate that IQSEC1 inhibition blocks growth and invasion in a multitude of in vitro 3D models across several cancer types, and tumour growth and metastasis in vivo. Elevated IQSEC1 expression is associated with clinical metrics of poor outcome: higher-grade tumours, metastasis, treatment resistance, PI3K pathway activation. Consequently, IQSEC1-ARF5/6 stratifies long-term recurrence-free survival in prostate cancer and more generally across a number of tumour types.

In conclusion, we identify a molecular mechanism that allows cells to determine whether to grow or invade in response to activation of an RTK: alternate isoforms of an effector protein, IQSEC1, differentially localise to modulate signalling to phosphoinositide signalling pathways.

## Methods

**Cell culture**. PC3 (ATCC), PC3 E-Cad+, TEM4–18, TEM2–5, GS689.Li, GS694. LAd, GS683.LALN, JD1203.Lu, GS672.Ug (M. Henry, University of Iowa), PC3-Epi and PC3-EMT (K. Pienta, Johns Hopkins School of Medicine) and LNCaP cells (H. Leung, Beatson Institute) were cultured in RPMI-1640 supplemented with 10% foetal bovine serum (FBS) and 6 mM L-glutamine. RWPE-1, RWPE-2, WPE-NB14 and CA-HPV-10 cell lines (ATCC) were grown in keratinocyte serum free media (K-SFM) supplemented with 50 μg/ml bovine pituitary extract (BPE) and 5 ng/ml epidermal growth factor (EGF). DU145 cells (ATCC) were maintained in minimum essential medium (MEM) supplemented with 10% FBS and 6 mM L-glutamine. VCaP, MDA-MB-231 (ATCC), murine KC Pten$^{fl/+58}$ (J. Morton, CRUK Beatson Institute) and TKCC-07 cells were cultured in Dulbecco's modified Eagle medium (DMEM) supplemented with 10% FBS and 6 mM L-glutamine. The TKCC-07 cells used in this project were provided by the Australian Pancreatic Cancer Genome Initiative (APGI) at the Garvan Institute of Medical Research (www.pancreaticcancer.net.au). 22Rv1 cells (H. Leung, Beatson Institute) were grown in phenol free RPMI-1640 containing 10% charcoal stripped FBS and 6 mM L-glutamine. HEK293-FT (Thermo Fisher Scientific) were cultured in DMEM with 10% FBS, 6 mM L-glutamine and 0.1 mM non-essential amino acids (NEAA) (all reagents from Thermo Fisher Scientific).

Growth factors or inhibitors were added as follows; 50 ng/ml hepatocyte growth factor (HGF) (PeproTech), 10 μM NAV-2729 (Glixx Laboratories), 10 μM QS11 (Tocris), 100 μM chloroquine (CST), 20 μM SecinH3 (Tocris), 25 μM cycloheximide (Sigma), 1 μM LY-294002 (Merck), 1 μM AZD8835 (AstraZeneca), 0.1 μM AZD8186 (AstraZeneca), 1 μM AS-6052-40 (Stratech), 1 μM Cal-101 (Stratech), and 10 μM AktII (Calbiochem) inhibitors.

Cells were routinely checked for mycoplasma contamination. PC3, RWPE-1 and RWPE-2 cells were authenticated using short tandem repeat (STR) profiling.

**Generation of stable cell lines**. Cell lines were made by co-transfecting plasmids with lentiviral packaging vectors (VSVG and SPAX2) into packaging cells (HEK293-FT) using Lipofectamine 2000 (Thermo Fisher Scientific). Viral supernatants were collected; filtered using PES 0.45 μm syringe filters (Starlab) to remove cell debris, and concentrated using Lenti-X Concentrator (Clontech) as per the

manufacturer's instructions. Cells were then transduced with the lentivirus for 3 days before either FACS sorting or selection with 2.5 μg/ml puromycin, 300 μg/ml G418 (both Thermo Fisher Scientific), 10 μg/ml blasticidin (InvivoGen) or 200 μg/ml hygromycin (Merck). Stable knockdown of proteins was achieved using pLKO.1-puromycin, pLKO.1-hygromycin or pLKO.1-membrane tagged Venus (substituted for puromycin) lentiviral shRNA vectors. ShRNA sequences are listed in Supplementary Table 3. GFP-IQSEC1 v2 and GST-GGA3-GAT were kind gifts from J. Casanova (University of Virginia). All RNAi-resistant variants and chimeras were made by mutagenesis or sub-cloning using fragment synthesis (GeneArt). GFP ARFs were kinds gift from P. Melancon (University of Alberta) and alternate fluorescent tags and mutations generated by sub-cloning. EGFP-PH-GRP1 and EGFP-PH-PLCδ were described previously[74] and sub-cloned into mNeonGreen. Myr-FLAG-Cre, Myr-FLAG-PIP5Ks, Myr-FLAG-PI3Ks, Myr-Akt1 and RFP-GGA1-NGAT were purchased from Addgene. LRP1-EGFP was a kind gift from S. Kins (Technical University Kaiserslautern).

**IQSEC1 variant information**. The nomenclature for IQSEC1 variants has been complicated by alternate names for IQSEC1 in literature and changing isoform designations at NCBI. Unification of nomenclature is presented in Supplementary Table 4.

**Live 3D culture and analysis**. Culture of cell lines as 3D acini was adapted from previous protocols[5]. Briefly, single cell suspensions were made ($1.5 \times 10^4$ cells per ml) in the appropriate medium supplemented with 2% growth factor reduced matrigel (GFRM; BD Biosciences). 150 μl of this mix was plated per well in a 96-well ImageLock plate (Essen Biosciences) pre-coated with 10 μl of GFRM for 15 min at 37 °C. Plates were incubated at 37 °C for 4 h, then imaged using an IncuCyte® ZOOM (Essen BioScience) with Incucyte ZOOM Live Cell Analysis System Software 2018A. Images were taken every hour for 4 days at 2 positions per well using a ×10 objective lens. Sample size (*n*) and replicate number are stated for each experiment in figure legends. Outlines of phase and GFP-positive (where appropriate) acini were generated using a custom pipeline in CellProfiler (Version 3.1.8). A custom macro in Fiji software (2.0.0) was then used to colour code images from each time point, progressively coloured along a blue-to-red rainbow timescale, and concatenate them into one image per 12-h block to reduce data dimensionality of multiday imaging. Using CellProfiler, measurements such as area and compactness, which could reliably measure size and protrusiveness of 3D PC3 objects, respectively, were generated for each 12-h block. Custom pipelines designed in KNIME Data Analytics Platform (Version 3.3.1) were then used to collate data from multiple experiments, normalise to controls, calculate Z-score and perform statistical analysis using one-way ANOVA. Normalised data and *p*-values are presented as heatmaps generated in PRISM 7 (GraphPad). The number of independent experiments (*n*), technical replicates, and the number of acini quantified per condition are stated in each figure legend.

**2D and 3D acini immunofluorescence and imaging**. 3D acini were set up as described above in either eight-well coverglass chamber slides (Nunc, LabTek-II) or in 96-well plates (Greiner) pre-coated with 60 or 10 μl GFRM respectively for 3 days. For 2D, cells were plated on a 96-well plate (Greiner) for 48 h prior to fixation. Samples were washed twice with PBS and fixed in 4% paraformaldehyde for 15 min. Samples were then blocked in PFS (0.7% fish skin gelatin/0.025% saponin/PBS) for 1 h at RT with gentle shaking and primary antibodies added overnight (1:100 unless stated otherwise) at 4 °C. After three washes in PFS, Alexa Fluor secondary antibodies (1:200), HCS CellMask™ Deep Red Stain (1:50,000) or Hoechst (1:1000) (all Thermo Fisher Scientific) were added for 45 min at room temperature (RT). Samples were maintained in PBS, after three 5 min washes in PBS, until imaging was carried out.

Cells or acini on chamber slides were imaged using a Zeiss 880 Laser Scanning Microscope with Airyscan. Images taken in super resolution mode on the Airyscan

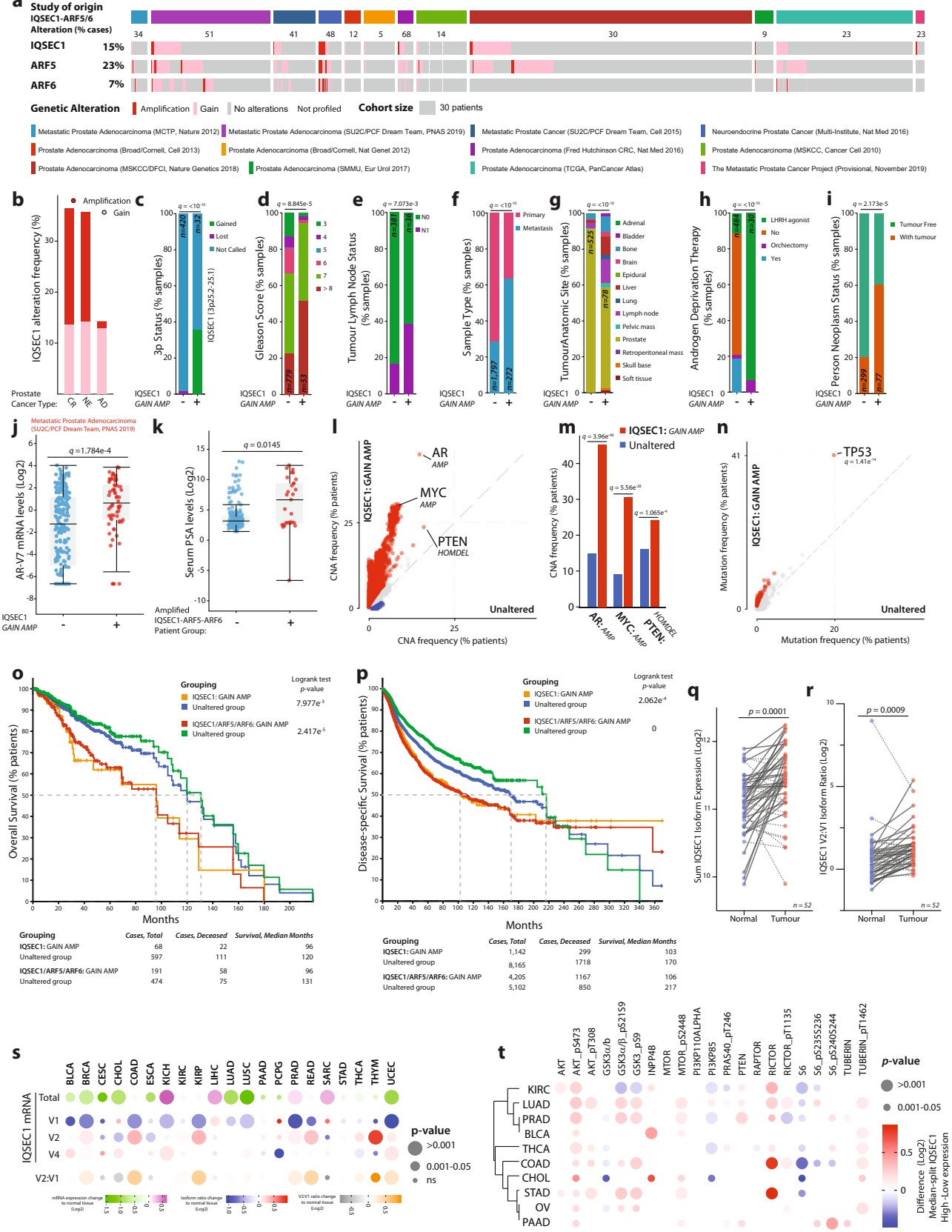

microscope were processed using the Zeiss proprietary ZEN 3.2 software, exported as TIFF files and processed in Fiji. Cells or acini on 96-well plates were imaged using an Opera Phenix™ high content analysis system and where appropriate Harmony High-Content Imaging and Analysis Software (PerkinElmer, Version 4.6) was used to perform machine learning.

Antibodies used: Alexa Fluor 488 (A12379), 568 (A12380) or 647 (A22287) phalloidin (Thermo Fisher Scientific), anti-IQSEC1 (Sigma, G4798), anti-IQSEC1 (Caltag-Medsystems, PSI-8009), anti-LRP1 (Sigma, L2295), anti-Met (CST, 3127), anti-Met phospho 1234/1235 (CST, 3077), anti-Akt phospho S473 (CST, 3787), DYKDDDDK Tag (CST, 14793, 1:800), anti-Ki67 (Thermo Fisher Scientific, 18-

**Fig. 8 IQSEC1 is associated with metastasis, treatment resistance, and poor clinical outcome. a** and **b** CN increase in *IQSEC1, ARF5, ARF6* across indicated cohorts. AD, prostate adenocarcinoma; NE neuroendocrine; CR Castrate-resistant. **c–j** Clinical metrics (% samples) between *IQSEC1* CN-amplified (+) or non-amplified group (−) for **c** 3p Status (−, $n = 420$; +, $n = 32$), **d** Gleason Score (−, $n = 779$; +, $n = 53$), **e** Tumour Lymph Node Status. N0, no positivity. N1, positivity. (−, $n = 381$; +, $n = 36$), **f** Sample Type (−, $n = 1,797$; +, $n = 272$), **g** Tumour Anatomic Site (−, $n = 525$; +, $n = 78$), **h** androgen deprivation therapy. Luteinizing hormone-releasing hormone, LHRH (−, $n = 484$; +, $n = 30$), **i** person neoplasm status (−, $n = 299$; +, $n = 77$), and **j** androgen receptor (*AR*) isoform v7 mRNA levels (Log2) (−, $n = 221$; +, $n = 69$). *p* values, **c–i** Chi-squared Test or **j** Kruskal–Wallis Test; *Q* values, Benjamini–Hochberg adjustment. Box-and-whiskers plots: 10–90 percentile; +, mean; dots, outliers; midline, median; boundaries, quartiles. **k** Prostate-specific antigen (PSA) serum levels (Log2) in *IQSEC1, ARF5, ARF6* CN increase (+, n = 22) or not (−, n = 86) patients. *Q* value, Kruskal–Wallis Test. Box-and-whiskers plots: 10–90 percentile; +, mean; dots, outliers; midline, median; boundaries, quartiles. **l–n** Copy Number Alteration (**l**, **m**) or (**n**) mutation frequencies between *IQSEC1* CN-amplified (n = 849) or unaltered (n = 2308) patients. Group association: Red, *IQSEC1* CN increase; blue, unaltered. *Q*-values, one-sided Fisher exact test with Benjamini–Hochberg adjustment. **o** and **p** Overall survival (% patients, months), unaltered compared to (i) IQSEC1 CN increase or (ii) combined *IQSEC1, ARF5, ARF6* CN increase for **o** cohorts in A (n = 665) or **p** pan-cancer TCGA cohorts (n = 9307). *p*-values, Log-rank *t*-test. **q** and **r** Matched Normal (blue) and prostate tumour (Tumour, red) Log2-normalised *IQSEC1* mRNA for **q** total isoforms or **r** v2-v1 ratio. values. Line: unbroken, tumour-increased; dotted, tumour-decreased. n = 52. *p*-values, Independent Groups *t*-test. **s** *IQSEC1* mRNA log2-normalised normal–tumour ratio across pan-cancer TCGA cohorts for total (green-magenta), individual isoforms (blue-red), or v2-v1 ratio (grey-orange). Circle size, *p*-value, Independent Groups *t*-test. Normal tissue (n = 643), tumour tissue (n = 6716). Supplementary Table 2 contains tumour-type data. **t** PI3K-AKT pathway (RPPA of pan-cancer TCGA cohorts) from median-split of Log2-transformed *IQSEC1* mRNA. Red, high *IQSEC1* co-occurring; blue, low *IQSEC1* co-occurring. Only cancer types presenting increased AKT_pS473 included. Row sorting from dendrogram. Circle size, *p*-value, Independent Groups *t*-test.

0192Z), anti-Cleaved Caspase 3 (CST, 9661), anti-PtdIns(3,4,5)P3 and anti-PtdIns (4,5)P2 (Echelon Biosciences, Z-P345B and Z-P045).

**2D and 3D phenotypic analysis of fixed samples**. Harmony High-Content Imaging and Analysis Software (PerkinElmer, Version 4.6) was utilised to perform machine learning using a number of custom designed pipelines. For 2D morphology assays the shape of each cell was defined by F-actin or HCS Cell-Mask staining and machine learning then used to classify cells as either spindle, spread or round phenotypes. Each cell was imaged in three consecutive planes (2 µm step size) and analysis was processed on each individual plane. Cells were detected based on nucleus localisation and defined by either F-actin or HCS CellMask staining. Cells in contact to the image border or without green channel positive signal were discarded. The morphology properties of each object were calculated to classify them into three different categories ('round', 'spindle' and 'spread') using machine learning following manual training. Different cell regions were defined as (a) 'peripheral', using Method resize region [µm/px], region type 'membrane region', Outer border: −2 px (−1.194 µm), Inner border: 5 px (2.985 µm), (b) 'nucleus', using method 'Standard' on nucleus region, (c) 'cytoplasm', using method 'restrict by mask': population 'green cells', mask regions 'Peripheral' and 'Nucleus'. Finally, intensity properties of PIP probe channel (Alexa 488) were calculated by standard method in the different cell regions within each object. As an alternate approach, images were also stack processed using maximum projection all slices, followed by a similar pipeline as described above. No significant difference was obtained between methods.

A custom pipeline was generated using KNIME Data Analytics Platform (Version 3.3.1) to collate data, calculate the log2 fold change of each phenotype over control and to calculate statistical significance using one-way ANOVA. Data obtained from three different replicates was collected and combined with experiment keys providing information about replicates and samples. For image analysis using single plane stack processing, objects that were in more than one plane were detected as a similar object using a similarity search of objects in subsequent planes based on euclidean distance search of nearest neighbour and distance selection using object X and Y positions. Upper bound (use range checking) used was 5.0. Automated selection of one single plane per object based on area (or alternatively, average of the three planes of the same object) was used for subsequent analysis. Selection of samples to be analysed, followed by selection of control sample, and log2 transformation of intensity data and normalisation to control was followed. In some cases this analysis was carried out in sub-populations of cells that either expressed a fluorescently tagged plasmid or were stained with a specific antibody.

The use of the nuclear, cytoskeletal and cytoplasmic reagents described above also allowed each cell to be segmented into specific sub-cellular regions i.e. nuclear, cytoplasmic and peripheral (defined as 5 pixels from outer edge of cell). When used in combination with fluorescently tagged proteins (such as reporters for phosphoinositides) or antibodies the mean intensity of a specific protein per cell (intensity/cell) or per sub-cellular region per cell (i.e. mean peripheral intensity/ mean total intensity/cell) could be measured. Where appropriate 'spots' of positive staining were detected within these sub-cellular regions to more accurately calculate the mean intensity/cell or mean spot area/cell.

Custom KNIME Data Analytics Platform (Version 3.3.1) pipelines described previously[74] were then used to collate data, to calculate the ratio over control and to determine statistical significance. In addition, these pipelines were adapted to measure total intensity of specific antibodies in 3D acini. Data are presented in box and whiskers plots as z-score normalised or as log2 fold change over control where each point represent one cell. The number of independent experiments (n),

technical replicates, number of cells/acini imaged per experiment, statistical test performed and significance is stated in the appropriate figure legend.

Harmony High-Content Imaging and Analysis Software (PerkinElmer, Version 4.6) was also used to quantify the colocalization between mNeonGreen-tagged ARF5 or ARF6 and RFP-GGA1-NGAT. PC3 cells stably expressing ARF/GGA1 were FACS sorted and a population that was positive for both mNeonGreen and RFP selected. These cells were plated for 24 h then treated with either NAV-2729 or QS11 overnight. Cells were then fixed, stained with CellMask (1:50,000) and Hoechst (1:1000) and imaged using an Opera Phenix™. A custom pipeline was created to identify red (GGA1+) and green (ARF+) 'spots' throughout each cell and to calculate the percentage of green spot area that overlapped with red spot area per cell. n = 2 with 4 replicates per condition with >500 cells imaged per condition.

**Animal studies**. Animal experiments were performed in compliance with all relevant ethical regulations and approvals of the relevant UK Home Office Project Licence (70/8645) and carried out with ethical approval from the Beatson Institute for Cancer Research and the University of Glasgow under the Animal (Scientific Procedures) Act 1986 and the EU directive 2010, and sanctioned by Local Ethical Review Process (University of Glasgow).

7-week-old CD1-nude male mice were obtained from Charles River (UK). Mice were housed at ambient temperature (19–22 °C), with relative humidity of 45–65%, and a 12 h–12 h light–dark cycle. $5 \times 10^6$ PC3 cells stably expressing either Scr (12 mice) or IQSEC1 KD4 (10 mice) shRNA were surgically implanted into one of the anterior prostate lobes of each mouse[75]. The mice were continually assessed for signs of tumour development and humanely sacrificed at an 8-week timepoint when tumour burden became restrictive. Primary tumours were imaged and tumour area and volume measured using VevoLAB ultrasound equipment and VevoLAB 3.1.1 software. Significance was calculated using Mann–Whitney, *$p \le 0.05$. Percentage of mice with tumours or with proximal (lumber lymph nodes and epididymal fat pads) or distal (thoracic lymph nodes and lungs) metastases were calculated by gross observation. *p* values were determined using a Chi-squared test (with Fisher's exact test) adjustment, n.s. = not significant, *$p \le 0.05$.

**Immunohistochemistry (IHC)**. 4 µm formalin fixed paraffin-embedded (FFPE) sections were cut from tissue blocks and maintained at 60 °C for 2 h. FFPE sections were stained with Cleaved Caspase 3 (CST, 9661) and Ki67 (Abcam, ab16667) antibodies using the Leica Bond Rx autostainer. Sections were loaded onto the autostainer, de-waxed (Leica, AR9222) and epitope retrieval carried out. Both antibodies were retrieved using ER2 buffer (Leica, AR640) for 30 min at 95 °C. Sections were rinsed with Leica wash buffer (Leica, AR9590) before peroxidase block was performed using an Intense R kit (Leica, DS9263). Caspase 3 was stained at a previously optimised dilution of 1/500 and Ki67 at 1/100 before washing with Leica wash buffer and then application of rabbit envision secondary (Agligent, K4003) and visualised using DAB in the Intense R kit. Sections were counterstained with Heamatoxylin and mounted using DPX (CellPath, SEA-1300-00A).

IHC slides were scanned using a Leica SCN 400F scanner at ×20 magnification and images uploaded to Halo Image analysis platform (Indica Labs.). Images were analysed using the CytoNuclear v1.5 algorithm. The percentage of Ki67 or Cleaved Caspase 3 positive cells in prostate tumours or lymph nodes was calculated. Additional measurements such as the average cell area and average cytoplasmic area were also taken.

**Met uptake assay**. Met uptake assays were performed using a fluorescently labelled Met-647 antibody (BD Biosciences) to assess the cells' ability to internalise and traffic Met. Cells plated on $5 \times 96$-well plates were equilibrated at 37 °C for 1 h in antibody binding medium (RPMI + 0.5% BSA + 1 M CaCl₂, pH 7.4). Cells were washed twice with ice-cold PBS, incubated with Met-647-conjugated antibody for 1 h at 4 °C (1:100) then washed again with PBS. One plate was fixed as described above. Two plates were treated with HGF (50 ng/μl) and chloroquine (100 μM) diluted in pre-warmed medium for 10 or 30 min at 37 °C. They were then acid stripped (washed three times at 4 °C with 0.5 M acetic acid, 0.5 M NaCl in PBS) and fixed as previously described. The remaining two plates were incubated at 17 °C with pre-warmed medium supplemented with HGF and chloroquine for 1 h. Acid stripping was performed and plates maintained at 37 °C without HGF or chloroquine for 10 min or 30 min prior to fixation. Cells were stained with Hoechst and phalloidin and imaged using the Opera Phenix™ High Content analysis system. Data are presented as line graphs that show normalised (Z-score) relative region intensities compared to the intensity of the whole cell. Box and whiskers plot shows total normalised (Z-score) intensity throughout the cells. $N = 2$ independent experiments, 4 replicates/condition with the following total number of cells examined per experiment: For internalisation at 0 min (Scramble 421, shIQSEC1 546), 10 min (Scramble 122, shIQSEC1 126), or 30 min (Scramble 93, shIQSEC1 120), and for recycling at 10 min (Scramble 150, shIQSEC1 136) or 30 min (Scramble 219, shIQSEC1 282) cells examined per experiment. p values were calculated using Welch's 2-tailed t-test and are shown on each graph as follows; n.s. = not significant, **$p ≤ 0.01$, ***$p ≤ 0.001$ and ****$p < 0.0001$.

**Protein purification**. pET20b-6xHis-ARF5D17 or pGEX-5x-1-hGGA3 [VHS-GAT] were transformed into BL21 (DE3) pLysS bacteria (Promega). Single colonies were grown in lysogeny broth (LB) medium in appropriate antibiotics until $OD_{600nm}$ of 0.3. Protein expression was induced using 200 μM isopropyl β-D-1-thiogalactopyranoside (IPTG, Sigma). Bacterial cells were pelleted and resuspended in 20 ml Buffer A (for His-tagged: 20 mM Tris, 300 mM NaCl, 5 mM MgCl₂, 150 μl BME; for GST-tagged: 20 mM Tris, 150 mM NaCl, 5 mM MgCl₂, 150 μl BME) containing protease inhibitors. DNAse (Sigma) was added to the lysate. Bacteria were lysed by passing through a 20,000 psi-pressurised microfluidizer. Lysates were collected at $20,000 \times g$ for 40 min at 4 °C and soluble proteins in the supernatant were sterile filtered 5 μm PVDF Membrane (Merck).

PCR products encoding human IQSEC1-SEC7-PH (residues 517-866 based on variant 2 numbering) WT and GEF-dead (E620K) were cloned into pEGFP-C1 fused with an N-terminal 12xHis tag followed by a 6× Glycine linker replacing the GFP sequence. HEK293 suspension cells were transfected with constructs using polyethylenimine (PEI, Sigma). HEK293 suspension cells were cultured at 37 °C and 8% CO₂ and 2×g. Cells were collected after 4 days, resuspended in His-IQSEC1 Buffer A (40 mM Tris 7.5 pH, 300 mM NaCl, 5 mM MgCl₂, 60 μl β-mercaptoethanol, 25 mM Imidazol) and sonicated. Soluble fraction was collected at 40 min, $21,000 \times g$ at 4 °C and sterile-filtered using a 0.45 μm PVDF-membrane (Merck).

Soluble fractions were loaded onto an ÄKTA purification system (GE Healthcare) using an equilibrated HisTrap or GSTrap (both GE Healthcare) column. The protein containing fraction was loaded onto the column at 1 ml/min. Proteins were eluted from columns using an imidozol gradient buffer (25–300 mM) or glutathione elution buffer (150 mM NaCl, 25 mM Tris, 20 mM glutathione) for His-tagged or GST-tagged proteins, respectively. Protein containing fractions were concentrated (Amigon Ultra-15 centrifuge filter) and gel purified on an appropriate size exclusion column (Superose-6 10–300 GL 24 ml for His-tagged IQSEC1, S75 16–600 120 ml for His-tagged ARF5, S200 16–600 120 ml for GST-GGA3) in gel filtration (GF) buffer (20 mM Tris, 150 mM NaCl, 1 mM DTT, 5 mM MgCl₂). Fractions containing purified proteins were concentrated, snap frozen and stored at −80 °C.

**Fluorescent polarisation assay**. Nucleotide exchange of ARF5 protein was examined by observing changes in fluorescent polarisation. 100 μM recombinant ARF protein was incubated with 200 μM mantGDP and 50 mM EDTA in gel filtration buffer (GF) (20 mM Tris, 150 mM NaCl, 1 mm DTT) overnight at 18 °C. 100 mM MgCl₂ was added to stop the exchange reaction. A PD10 (GE Healthcare) desalting column was then washed with GF buffer containing 5 mM MgCl₂, 500 μl of sample added and then eluted with GF buffer. Protein concentration was determined using a Bradford assay as per manufacturers instructions. 20 μM of GDP nucleotide and 2 μM of GEF protein was sequentially added to 1 μM ARF and polarisation changes measured with a Photon Multiplier Detection System (Photon International Technology). Excitation was set to 366 nm and emission to 450 nm. Inhibitors were added to the reaction prior to addition of GEF protein.

**Proliferation assays**. PC3 cells were plated in a 96-well ImageLock plate, in triplicate for 24 h. Imaging was carried out on IncuCyte® ZOOM each hour for 48 h. Cell area per well was then measured using the IncuCyte® ZOOM analysis software. $n = 3$ independent experiments with three replicates per condition. p-values (Student's t-test): **$p ≤ 0.01$ and ****$p ≤ 0.0001$.

**Invasion and migration assays**. ImageLock plates were coated with 10% GFRM diluted in medium overnight at 37 °C. Cells were re-suspended in 100 μl medium and plated for 4 h at 37 °C. The resultant monolayer was wounded using a wound making tool (Essen Biosciences), washed twice with medium to remove debris, and overlaid with either 50 μl of 25% GFRM for invasion assays or 100 μl medium for migration assays. After an hour at 37 °C 100 μl medium was added to invasion assays and plates imaged every hour for 4 days using the IncuCyte® ZOOM. The number of independent experiments ($n$) and technical replicates per experiment is stated in the appropriate figure legend. Results are presented as relative wound density (RWD) for each time point and for the time point at which the average RWD of the control samples is 50% ($T_{max}1/2$). Values are mean ± s.d., p values were calculated using a Student's 2-tailed t-test and are shown on each graph as follows; n.s. = not significant, *$p ≤ 0.05$, **$p ≤ 0.01$, ***$p ≤ 0.001$ and ****$p < 0.0001$.

**Immunoblotting**. Cells were plated for 24 h prior to treatment with growth factors or inhibitors for a further 24 h. Plates were washed twice with ice cold PBS then lysis buffer added for 15 min (50 mM Tris–HCl, pH 7.4, 150 mM NaCl, 0.5 mM MgCl₂, 0.2 mM EGTA, and 1% Triton X-100 with cOmplete protease inhibitor cocktail and PhosSTOP tablets (Roche). Cells were scraped and lysates clarified by centrifugation at $216 \times g$ at 4 °C for 15 min. BCA Protein Assay kit (Pierce) was used to determine protein concentration. SDS–PAGE was then performed and proteins transferred to PVDF membranes using the iBlot 2 transfer system (Thermo Fisher Scientific). Membranes were incubated for an hour in Rockland blocking buffer (Rockland) and primary antibodies added overnight at 4 °C (1:1000 unless stated otherwise). Antibodies used were as follows: anti-GAPDH (CST 2118 1:5000), anti-IQSEC1 (Sigma G4798), anti-IQSEC1 (Caltag-Medsystems PSI-8009), anti-GFP (Merck 000000011814460001), anti-LRP1 (Sigma L2295), anti-Met (CST 3127), anti-Met phospho 1234/1235 (CST 3077), anti-Akt (CST 2920), anti-Akt phospho S473 (CST 3787), anti-ARF1 (Novus Biologicals NB-110-85530), anti-ARF6 (Sigma A5230), ARF5 (Novus Biologicals H00000381-M01), anti-ARFGAP1 (Sigma HPA051019), anti-SORL1 (BD 611860), anti-Sin1 (CST 12860), anti-RICTOR (CST 2114), anti-PPC6 (Sigma HPA050940 1:250), anti-14-3-3ζ/Δ (CST 7413), anti-PIP5Ka (CST 9693) and PIP5Kb (Sigma K0767). After addition of appropriate secondary antibodies for 45 min, membranes were washed three times in TBST and imaged using a ChemiDoc Imager (BioRad) or Odyssey Imaging System (LI-COR Biosciences). Bands were quantified using Image Lab 6.1 (BioRad) or Image Studio Software 6.0 (LI-COR Biosciences). The number of independent experiments ($n$) is stated in the appropriate figure legend and quantitation is shown as mean ± s.d., p values were calculated using a Student 2-tailed t-test, unless otherwise stated, and are shown on each graph as follows; n.s. = not significant, *$p ≤ 0.05$, **$p ≤ 0.01$, ***$p ≤ 0.001$ and ****$p < 0.0001$. GAPDH was used as a loading control for each immunoblot and a representative image for each sample set is shown where appropriate.

**GFP Trap and Immunoprecipitation**. Immune complexes were collected when 1 mg of cell lysate was immunoprecipitated with anti-Met antibody (CST 3127, 1:50) overnight at 4 °C with rotation. Anti-mouse agarose or mouse agarose (both Sigma) were added for 1 h at 4 °C prior to three washes in lysis buffer. Samples were then separated by SDS–PAGE, transferred to a PVDF membrane and immunoblotted. Lysates from cells expressing GFP-tagged proteins were immunoprecipitated using a GFP-Trap Kit (Chromotek) as per manufacturer's instructions and immunoblotting performed as above. $n = 3$ and quantitation is shown as mean ± s.d.

**Mass spectrometry**. For mass spectrometry analysis agarose beads were resuspended in a 2 M urea and 100 mM ammonium bicarbonate buffer and stored at −20 °C. On-bead digestion was performed from the supernatants. Triplicate biological replicates were digested with 25 μl 2 M urea in 50 mM Tris, pH 7.5, 1 mM DTT, and 150 ng EndoLysC (Alpha Laboratories) and 150 ng trypsin (Promega) on beads Lys-C (Alpha Laboratories) and trypsin (Promega) on beads as according to optimised approaches[76]. Tryptic peptides were separated on a 20 cm fused silica emitter (New Objective) packed in house with reverse phase Reprosil Pur Basic 1.9 μm (Dr. Maisch GmbH) using an EASY-nLC 1200 (Thermo Fisher Scientific) coupled online to an Orbitrap Q-Exactive HF mass spectrometer (Thermo Fisher Scientific) via nanoelectrospray ion source (Thermo Fisher Scientific). For the full scan a resolution of 60,000 at 250Th was used. The top ten most intense ions in the full MS were isolated for fragmentation with a target of 50,000 ions at a resolution of 15,000 at 250Th. MS data acquisition were performed using the XCalibur software (Thermo Fisher Scientific). The MaxQuant software version 1.5.5.1[77] was used to process MS Raw files and searched with Andromeda search engine[78], querying UniProt[79]. Database was searched requiring specificity for trypsin cleavage and allowing maximum two missed cleavages. Methionine oxidation and N-terminal acetylation were specified as variable modifications, and Cysteine carbamidomethylation as fixed modification. The peptide and protein false discovery rate (FDR) was set to 1%. The common reverse and contaminant hits (as defined in MaxQuant output) were removed. Only protein groups identified with at least one uniquely assigned peptide were used for quantification. For label-free quantitation, proteins quantified in all three replicates in at least one group, were measured

according to the label-free quantitation algorithm available in MaxQuant[80]. Significantly enriched proteins were selected using a Welch *t*-test with a 5% FDR (permutation based). Hits were prioritised based on fold change expression data mined from publicly available RNAseq data from PC3 sublines from published data sets (PC3E versus GS689.Li; SRS354082 [https://www.ncbi.nlm.nih.gov/sra/?term=SRS354082]) and PC3-Epi versus PC3-EMT14; GSE48230 [https://www.ncbi.nlm.nih.gov/geo/query/acc.cgi?acc=GSE48230][39,40].

**GGA3 pulldown.** A pulldown assay was performed to examine GTP-loading of ARF proteins using the ARF-GTP-specific binding domain of GGA3. Cells were cultured for 48 h then serum starved overnight. HGF in serum free medium was added for 30 min. Cells were then lysed on ice in pulldown-lysis buffer (50 mM Tris, 100 mM NaCl, 2 mM MgCl$_2$, 0.1% SDS, 0.5% Na-deoxycholate, 1% Triton X-100, 10% glycerol), syringed five times and centrifuged at 4 °C 14,000×*g* for 1 min. Spin columns were equilibrated with 50 µl of glutathione agarose resin and washed with pulldown-column wash buffer (1:1 pulldown-lysis buffer and 1×TBS). 80 µg of GST-GGA3-GAT recombinant fusion protein was immobilised on the agarose resin by incubation at 4 °C with gentle rocking for 1 h. 1.5 mg of lysate was added onto spin columns and incubated at 4 °C for 2 h with rocking. Unbound proteins were washed off the column with pulldown-wash buffer (50 mM Tris, 100 mM NaCl, 2 mM MgCl$_2$, 1% NP-40, 10% glycerol). 60 µl of pulldown-elution buffer (10 mM glutathione in 1×TBS) was added to the spin column and incubated for 5 min at RT. Eluted protein was collected at 1250×*g* for 1 min and samples were prepared for SDS–PAGE and western blotting as described above.

**qPCR.** RNA was extracted using an RNeasy kit (Qiagen) and reverse transcription of RNA performed using a High-Capacity cDNA Reverse Transcription Kit (Thermo Fisher Scientific) following the manufacturer's protocols. TaqMan qPCR was then carried out as per manufacturer's instructions (Thermo Fisher Scientific). $n = 3$ technical replicates. Data were analysed using the Applied Biosystems 7500 Software v2.0.6 and the relative quantitation (RQ) was calculated using the comparative $C_t$ ($\Delta\Delta C_t$) method. ARF and ARFGEF expressions were detected using a custom qPCR primer panel. Details are available on request.

**Analysis of patient cohorts.** The majority of patient data (Copy number, mutational status, RNAseq, RPPA, co-expression matrices and clinical annotation) was accessed, analysed and downloaded using in-platform cBioportal.org tools[81,82]. Normal versus Tumour RNAseq, including transcript variant expression, was obtained by downloading IQSEC1 variant annotation across the TCGA pan cancer dataset using the TCGA Splicing Variants Database (www.TSVdb.com)[83]. RPPA and IQSEC1 variant data were analysed using custom KNIME Data Analytics Platform (Version 3.3.1) pipelines.

**Reporting summary.** Further information on experimental design is available in the Nature Research Reporting Summary linked to this paper.

## Data availability

The proteomic data generated in this work have been deposited to the ProteomeXchange Consortium via the PRIDE partner repository[84] with the dataset identifier PXD013810. The RNAseq data from PC3 sublines in this study are available in either the Short Read Archive database for PC3E, GS689.Li in SRS354082), or the Gene Expression Omnibus for PC3-Epi, PC3-EMT14 in GSE48230. Any other data that supports the findings of this study are available from the corresponding author upon reasonable request. Source data are provided with this paper.

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

## Acknowledgements

This work was supported by the following grants; D.M.B. NIH K99CA163535, M.N. CRUK (C596/A19481), K.N. CRUK (C7932/A25170), E.S., A.R.-F, L.M., S.Z., and S.L. CRUK C596/A17196, E.M. CRUK A25142, S.I. and T.Y. CRUK (A19257). E.C.F. was supported by a University of Glasgow Industrial Partnership Ph.D. scheme co-funded by Essen Bioscience, Sartorius Group. We would like to thank the Core Services and Advanced Technologies at the Cancer Research UK Beatson Institute, with particular thanks to the Beatson Advanced Imaging Resource, Histology and Molecular Technologies. We thank Wenjie Sun for assistance with the TCGA Splicing Variants Database.

## Author contributions

M.N., D.M.B., K.N., E.S. designed experiments and analysed data. M.N., D.M.B., K.N., E.S., T.Y., A.R.-F., R.P., L.G. and S.M. performed experiments and analysis. L.M., E. Shanks and E.C.F. helped with the development of high-throughput imaging and analysis. S.L. and S.Z. performed and analysed the proteomic experiments. T.Y. and S.I. guided protein purification and biochemistry. D.M.B. and E.M. analysed patient data. S.M., R.P., and L.G. performed and analysed, while K.B. and H.L. guided in vivo experiments. J.P.M. generated and/or provided cell lines. D.M.B., M.N. and E.S. wrote the manuscript. D.M.B. supervised the study. All authors discussed the study and commented on the manuscript.

## Competing interests

E.C.F. was supported by a University of Glasgow Industrial Partnership Ph.D. scheme co-funded by Essen Bioscience, Sartorius Group. All other authors have no competing interests.

**Additional information**

