## [Peer Review File · Nature Communications]

Reviewers' comments:

Reviewer #1 (Remarks to the Author); expert in ARF GTP:

This manuscript describes a novel and important role for the ARF-specific guanine nucleotide exchange factor (GEF) IQSec1 in the growth and invasive capacity of prostate cancers. The study is a tour-de-force, defining a specific IQSec1 splice variant that promotes invasiveness as well as an interactome for each of the expressed variants, defining the roles of a subset of interacting proteins in growth vs. invasiveness, and the requirement for IQSec1 in each of those roles. They define a signaling pathway leading from IQSec1 through ARF5 and ARF6 to PI5- and PI3 kinases to Akt that promotes invasion in conjunction with the RTK Met and a second receptor LRP1. Finally, the authors demonstrate convincingly that IQSec1 is important for the metastasis of prostate cancer in a mouse model. The study is well designed, incredibly thorough, and contains a wealth of information that will be of use to workers in the field. In fact, if anything it contains too much information, making it difficult to follow in places.

With a few exceptions the data are clear, convincing and support the conclusions drawn by the authors. Specific issues that remain to be addressed:

1. Although it is clear that IQSec1 is important for the activities of Met and Akt, the experiments that attempt to define mechanism are a weak point in the study. They are based almost entirely on imaging, and in many cases the observed changes +/- IQSec1 are small. This reviewer is not convinced that loss of IQSec1 significantly alters the distribution of either Met or pAkt (Fig 5D, S6L). Some of the immunoblot data is also confusing; in Fig. S5A, knockdown of IQSec1 significantly attenuates activation of Met by HGF, but has no obvious effect on Aktp473. Since Akt is activated downstream of Met, why are these events uncoupled here?
2. Similarly, the changes in phosphoinositide levels are based entirely on imaging and are also subtle, at least in 2D. The distribution is clear in control 3D cultures (Fig. 5B), but since projections don't form in the absence of IQSec1 it is impossible to compare controls to IQSec1-deficient cells. While it is plausible that attenuated PIP2 and/or PIP3 synthesis would impair protrusion formation, the data as shown are not conclusive and the authors should tone down their interpretation.
3. Fig. 6I, J shows that even membrane anchored (myristoylated) Akt is not sufficient to support outgrowth of cells from aggregates in the absence of IQSec1. The authors suggest that this may be due to Akt-dependent events that don't require S473 phosphorylation, but what would those be? The suggestion that it is due to spatial control of the signals is only marginally supported by the data. It seems equally likely that IQSec1 has Akt-independent activities that are necessary but not sufficient, and that Akt is also necessary but not sufficient in the absence of IQSec1. For example, ARFGAP1 appears to be acting as an Arf5/6 effector and is essential for invasiveness – does its function have anything to do with Akt?
4. A minor point - the authors report that PC3 cells express 4 splice variants of IQSec1 which they name v1-4, however they do not provide accession numbers or sequence information to identify these variants in Genbank or other databases. Because the content of the N-terminus and C-terminus are important to function, it is important to know what those exons look like at the sequence level.

Reviewer #2 (Remarks to the Author); expert in imaging:

This is a very detailed investigation of the role of a GEF exchange factor for the Arf GTPases, IQSEC. The volume of work performed is very impressive and it is very clear that IQSEC has several important functions, but it is not clear that all of the effects of IQSEC knockdown are related and some of the stated conclusions do not seem to be justified. Most importantly, despite the prominence of this claim in the title and highlights, there is no evidence that IQSEC is restricting PIP3 to protrusion tips and the observation that PIP3 is enriched in cellular protrusions is well established. It seems possible that IQSEC knockdown simply reduces phospholipid concentrations overall and one of the consequences of this is reduced invasion.

I have several other more specific critiques:

1. In figure 2J, the authors claim that IQSEC1v2 localizes to a discrete pool in protrusions, but the image simply shows some enrichment in one protrusion tip, while there is quite a bit of staining in the rest of the cell.
2. I am not convinced that the V2 transcript is completely cytoplasmic. Image in Figure 2 shows extremely heterogeneous expression of GFP-IQSEC1V2 and the image is saturated in more highly expressing cells. Accounting for this expression heterogeneity, the localization of V2 looks similar to V4.
3. The quantification of phospholipid localization is not very rigorous, because use of the PH-domain biosensors to quantify relative PIP levels requires the ability to measure only membrane-localized PH domains. It is not clear to me exactly how these measurements were performed, but simply using phalloidin staining to identify the cortex is not sufficient because it does not exclude PH probe that is proximal to the actin cytoskeleton but is not membrane-localized. Depending on the exact resolution of the microscope used, there could be significant overlap between a phalloidin image and a PH probe image that is not actually labeling membrane-localized PH probe. This is especially important given that the authors are comparing different cells populations that may have different membrane morphologies that would perturb such measurements.
4. The conclusion that specific pools of Arf5/6 are critical seems to be based mostly on the result that expression of the fast cycling mutants does not rescue IQSEC depletion. Do the fast cycling mutants increase the GTP loading, or is Arf5/6 saturated under these conditions? Is there a constitutively active mutant that can be tested?
5. Why is Akt ppn used as the readout of PI3K inhibitor? What is the positive control for PI3K inhibitors efficacy? The PH-domain biosensors may be a more precise readout of PI3K inhibitor activity.
6. How was the pS473 spot area measured in Figure S6M&N? It looks like from the images that the increase in spot area may be due to overlapping spots, possibly due to changes in cell morphology because of IQSEC knockdown. It's also not clear how the statistics were calculated. How many cells were analyzed? Were the statistics calculated based on each spot being an independent measurement? In the case of image analysis of subcellular structures, it is not permissible to consider each measurement to be statistically independent if they are all taken from the same cell, because one or two cells with a very strong phenotype measured many times will strongly bias the results.
7. How many mice were included in the cohort for Figure 7B? 7D? What are the p values? The lack of error bars suggests insufficient power for this experiment. The zero size of many tumors in Panel E suggest that there is an engraftment defect due to IQSEC knockdown. Furthermore, the authors claim that tumor size suggests a proliferation defect, but I disagree with this inference. If the authors want to make any conclusion regarding proliferation in these xenograft experiments, then they will need to measure proliferation and not tumor size, which is affected by the balance between

proliferation, apoptosis, and cell movement.

8. There is no evidence that the lack of macromets is due to invasion, as it could be due to survival or proliferation defects, both of which are required for metastatic seeding and outgrowth into macromets.

9. It is not clear that IQSEC1 specifically depletes ARF GTP-binding in protrusions. Although it's not clear to me how the % overlap analysis was performed, a simple decrease in GTP-loading due to IQSEC knockdown would cause a decrease in colocalization overall and not just in protrusion tips. Furthermore, there are regions of GTP-bound ARF in Figure 3J that are not in the protrusion tips.

10. Recovery of the invasion phenotype due to expression of the myristoylated PI3Kbeta looks relatively mild although I can't tell for sure based on how the data is presented. Thus it seems plausible that other effects of IQSEC knockdown might be causing the invasion defect.

Reviewer #3 (Remarks to the Author); expert in prostate cancer and mouse/organoids models:

Manuscript NCOMMS-19-539476

Major and overlapping signaling pathways are activated in prostate cancer to drive pleiotropic effects on cancer phenotype, including the stimulation of growth and invasion. However, prostate cancer progression proceeds through temporally separated stages with growth dysregulation typically preceding invasion. This application addresses an unresolved fundamental question, how is this signaling decoded to elicit differential effects on growth and invasion at different times and places. The data presented suggest one such mechanism, expression of IQSEC1 (a GTPase exchange factor) transcript isoforms that restrict PI(3,4,5)P3 production to discrete domains resulting in invasion-driving cellular protrusions.

This is a detailed mechanistic study whose data link several steps in the proposed signaling cascade, making a compelling case that spatially restricted, RTK-driven, IQSEC1 dependent, PI metabolism generates cellular protrusions that drive cancer cell invasion. While the general concept that spatially restricted signaling is important for migration/invasion is not novel, the particular IQSEC1 dependent mechanism described is novel in prostate cancer. Another interesting finding is that IQSEC1 is important not only for local AKT activation that drives invasion, but it is also required for trafficking of pAKT to other regions (juxtannuclear) where it drives overall cell growth.

The limitations of the current manuscript largely revolve around significance. All the experiments rely on established prostate cancer cell lines, almost exclusively the PC3 cell line. The potential limitation here is the mechanism described may be idiosyncratic in the PC3 line. There is some validation using other cell lines in supplementary figure 7, but this is restricted to use of a pharmacological ARF6 inhibitor. The manuscript would be strengthened by validation of the IQSEC1 mediated mechanism (genetic, imaging) in additional prostate cancer cell lines. Even better would be validation in more physiological experimental models (patient derived organoids, xenografts, etc.). Additionally, all the IQSEC1 isoforms drive cell growth, with some having a greater effect on invasion. Hence effects on growth and invasion have not been genetically separated cleanly. This makes it difficult to ascribe the effects of IQSEC1 knockdown on metastasis in figure 7 to the mechanism described, rather than to general effects on tumor cell growth. The hypothesis predicts it should be possible to generate PC3 cells that grow robustly but metastasize weakly by removing specifically the invasion promoting IQSEC1 isoform; this should be tested if feasible.

Technical issues to be resolved:

- 1) I am not sure “spatially restricted” is the most accurate term to describe the mechanism since IQSEC1 is also important for trafficking signals elsewhere; is “spatially concentrated” signaling more accurate?
- 2) The authors may want to speculate on what the spatially restricted signaling is doing to drive formation of protrusions. Is it local macromolecular synthesis?
- 3) The authors make the assumption that their morphological measure (i.e. area, compactness, protrusions) accurately predicts physiological invasion. This assumption should be validated in a manner more rigorous than wound healing assays (i.e. Boyden chamber assays, metastasis in vivo, etc.).
- 4) The data presented in figure 2J showing highly localized IQSEC1v2 immunostaining in the protrusion tip is compelling, but there is no quantitative measure of what fraction of protrusions show this immunostaining pattern. This should be stated or shown. If other isoform specific antibodies are available, it would be important to compare them in this assay.
- 5) It is not clear why the clinical correlations presented in figure 1J utilize a TCGA dataset (n=487) while the clinical correlations presented in figure 7M use a different, smaller dataset (n=79). Why were these particular data sets chosen? The manuscript would be strengthened by analysis of multiple, independent publically available datasets inform the reader how general the findings are.
- 6) The authors may want to comment on whether this mechanism is unique to prostate cancer or can be generalized to other cancers.

Overall, this manuscript would likely fill fundamental knowledge gaps and be of substantial interest to researchers in the field if revised to account for these limitations and technical issues.

David W. Goodrich, Ph.D.

Response to Reviewers

We thank the reviewers for their insightful comments. We respond to all comments with experimental work or clarification. We now demonstrate a generalised requirement for IQSEC1 in a collection of additional 3D models, and provide extensive clinical and genomic characterisation of IQSEC1 across multiple tumour types. This revealed a robust association of IQSEC1 levels with phosphoinositide signalling, the occurrence of IQSEC1 isoform to the pro-invasive variant in multiple tumour types, and an association of IQSEC1 with metastasis and poor clinical outcome in large numbers of patients. We feel this strengthens identification of IQSEC1 as a regulator of phosphoinositide signalling that promotes metastasis. We have highlighted these changes in the revised manuscript (in blue).

Reviewers' comments:

Reviewer #1 (Remarks to the Author); expert in ARF GTP:

This manuscript describes a novel and important role for the ARF-specific guanine nucleotide exchange factor (GEF) IQSec1 in the growth and invasive capacity of prostate cancers. The study is a tour-de-force, defining a specific IQSec1 splice variant that promotes invasiveness as well as an interactome for each of the expressed variants, defining the roles of a subset of interacting proteins in growth vs. invasiveness, and the requirement for IQSec1 in each of those roles. They define a signaling pathway leading from IQSec1 through ARF5 and ARF6 to PI5- and PI3 kinases to Akt that promotes invasion in conjunction with the RTK Met and a second receptor LRP1. Finally, the authors demonstrate convincingly that IQSec1 is important for the metastasis of prostate cancer in a mouse model. The study is well designed, incredibly thorough, and contains a wealth of information that will be of use to workers in the field. In fact, if anything it contains too much information, making it difficult to follow in places.

With a few exceptions the data are clear, convincing and support the conclusions drawn by the authors. Specific issues that remain to be addressed:

Thank you for your positive characterisation of our work. We have attempted to balance the addition of more work in this revision with streamlining some data to simplify the manuscript.

1. Although it is clear that IQSec1 is important for the activities of Met and Akt, the experiments that attempt to define mechanism are a weak point in the study. They are based almost entirely on imaging, and in many cases the observed changes +/- IQSec1 are small. This reviewer is not convinced that loss of IQSec1 significantly alters the distribution of either Met or pAkt (Fig 5D, S6L).

We see imaging as a strength that compliments biochemical characterisation as it considers spatial regulation.

Our high-content imaging approaches allowed imaging >1,000 cells per condition, with each cell segmented into cortical, cytoplasmic and juxtannuclear regions. This allows fine-grained, robust statistical analysis of discrete Met and Akt localisation to a scale not previously reported. The end result of this nuanced imaging approach to divide cells into multiple regions is that changes are subtle, but robustly significant: IQSEC1 knockdown slows both Met and Akt trafficking to the juxtannuclear region (Fig 5, S8H-K). This was corroborated by biochemical characterisation showing IQSEC1 knockdown slows Met turnover (Fig S6E).

While we stand by the robustness of the data and methods, we do agree that the effect of IQSEC1 on Met/Akt localisation is subtle and not all-or-none. This contrasts with the strong alteration in phenotype upon IQSEC1 loss. Accordingly, we focus on signalling effects of IQSEC1 and toned down our emphasis on Met trafficking as a major mechanism. We change the results section characterising trafficking to the title 'An IQSEC1-LRP1 complex modestly regulates Met endocytic trafficking' and state at the end of that section:

"The magnitude of effects of IQSEC1 depletion on trafficking, however, were modest. We therefore turned to potential effects on IQSEC1 in regulating signalling from Met."

As a note, we attempted imaging of fluorescently tagged Met and Akt but were precluded by technical limitations. GFP-Akt1 displayed mostly cytoplasmic labelling. We were unable to make Met-GFP lines, a well-known problem in the Met field (Met overexpression induces ligand-independent de-adhesion causing cells to float off substrate; PMID: 24790222 and Stephanie Kermorgant, UCL, expert in Met signalling, personal communication). We instead provide new high-resolution immunofluorescence images of endogenous pMet and pAkt (Fig 5H, S8H).

Some of the immunoblot data is also confusing; in Fig. S5A, knockdown of IQSec1 significantly attenuates activation of Met by HGF, but has no obvious effect on Aktp473. Since Akt is activated downstream of Met, why are these events uncoupled here?

We repeated western blot characterisation of Met and Akt (total and phospho) +/- IQSEC1 with seven more replicates (Fig S5A-B, S6A-B). This overcame some previous variability that lead to a lack of significance. We now demonstrate clear concordance between Met and Akt activation.

2. Similarly, the changes in phosphoinositide levels are based entirely on imaging and are also subtle, at least in 2D. The distribution is clear in control 3D cultures (Fig. 5B), but since projections don't form in the absence of IQSec1 it is impossible to compare controls to IQSec1-deficient cells. While it is plausible that attenuated PIP2 and/or PIP3 synthesis would impair protrusion formation, the data as shown are not conclusive and the authors should tone down their interpretation.

This point is somewhat shared with Reviewer 2.

We counter that the effects of IQSEC1 on phosphoinositide distribution cannot be described as subtle without a reference point. We provide this context and demonstrate that IQSEC1 effects are at the same magnitude as the strongest known inhibitor of PIP3 pathways: pan-inhibition of Class I PI3-kinases (new Fig S7F).

We also now demonstrate that IQSEC1 is required for cortical PIP2/PIP3 localisation to form a protrusion and **not** total levels. IQSEC1 depletion actually leads to an *increase* in total PIP2/PIP3 (Fig S7J), but these phosphoinositides become delocalised to intracellular compartments (Fig S7A). Effects of IQSEC1 on PIP2/PIP3 are therefore overt and related to their cortical enrichment, not total synthesis.

3. Fig. 6I, J shows that even membrane anchored (myristoylated) Akt is not sufficient to support outgrowth of cells from aggregates in the absence of IQSec1. The authors suggest that this may be due to Akt-dependent events that don't require S473 phosphorylation, but what would those be? The suggestion that it is due to spatial control of the signals is only marginally supported by the data. It seems equally likely that IQSec1 has Akt-independent activities that are necessary but not sufficient, and that Akt is also necessary but not sufficient in the absence of IQSec1.

The role of Akt is complex. While Akt inhibition blocked both growth and invasion (Fig S8B,C), Myr-Akt is not sufficient to promote invasion even in the presence of IQSEC1. Our experience in 3D culture is that agents that strongly block growth mostly also block invasion. This suggests that following PIP3 generation the signal transduction event branches: Akt-dependent growth (which is essential to develop to the stage to make protrusions) and Akt-independent, PIP3-dependent invasion.

Importantly, IQSEC1 depletion also blocked myr-Akt-dependent growth promotion even though - as the reviewer points out - biochemical markers of Akt activation (pS473 levels) would suggest it is active. We identify that IQSEC1 controls localisation of Akt. We provide streamlined quantitation and higher resolution images of pAkt localisation and levels upon IQSEC1 KD across thousands of cells. IQSEC1 KD decreases total pAkt level (S6AB), resulting in lack of juxtannuclear pAkt seen in controls (S8H), instead accumulating pAkt in larger patches (Fig S8J) of less intensity (Fig S8I) at the cell cortex.

Why exactly Akt needs to be in a juxtannuclear region remains to be elucidated, but likely involves access of Akt to phosphorylation substrates that may reside in distinct membrane compartments.

For example, ARFGAP1 appears to be acting as an Arf5/6 effector and is essential for invasiveness - does its function have anything to do with Akt?

ARFGAP1 depletion does lower pS473-Akt levels (Fig S5N), suggesting ARFGAP1 may be part of the same TORC2 regulatory mechanism as IQSEC1-ARF5/6. As this reviewer pointed out, this work is already very dense. We argue the mechanisms of how ARFGAP1 functions in this module will need to wait for an independent study.

4. A minor point - the authors report that PC3 cells express 4 splice variants of IQSec1 which they name v1-4, however they do not provide accession numbers or sequence information to identify these variants in Genbank or other databases. Because the content of the N-terminus and C-terminus are important to function, it is important to know what those exons look like at the sequence level.

This is a great point. We provide Table 1 to clarify the nomenclature for these variants. Also see Fig 2A for illustration.

Reviewer #2 (Remarks to the Author); expert in imaging:

This is a very detailed investigation of the role of a GEF exchange factor for the Arf GTPases, IQSEC. The volume of work performed is very impressive and it is very clear that IQSEC has several important functions, but it is not clear that all of the effects of IQSEC knockdown are related and some of the stated conclusions do not seem to be justified. Most importantly, despite the prominence of this claim in the title and highlights, there is no evidence that IQSEC is restricting PIP3 to protrusion tips and the observation that PIP3 is enriched in cellular protrusions is well established. It seems possible that IQSEC knockdown simply reduces phospholipid concentrations overall and one of the consequences of this is reduced invasion.

This is a great point that helped us to clarify our mechanism. Our new data supports our original assertion: IQSEC1 is not required for general PIP2/PIP3 synthesis, but instead controls PIP2/PIP3 restriction to the cell cortex. See response to Reviewer 1, point 2 for full explanation and Fig S7 for data. We also emphasise that although PIP3 in single cell invasion has been localised to protrusions, this is the first demonstration that cells coordinate this polarity across a multicellular scale to promote collective cell invasion.

I have several other more specific critiques:

1. In figure 2J, the authors claim that IQSEC1v2 localizes to a discrete pool in protrusions, but the image simply shows some enrichment in one protrusion tip, while there is quite a bit of staining in the rest of the cell.

We now provide clarification that IQSEC1 v2 is also cytoplasmic, added images of IQSEC1 v2 in multiple protrusions that occur from a cyst, and added quantitation of the prevalence of IQSEC1 in protrusion tips (on page 8):

“Accordingly, anti-IQSEC1 antibodies directed to either all isoforms or to v2-specifically (Fig 2H) labelled both the cytoplasm and tubulovesicular compartments behind the tips of invasive acinar protrusions in 3D (Fig 2I-K, arrowheads, S2J, K).”

2. I am not convinced that the V2 transcript is completely cytoplasmic. Image in Figure 2 shows extremely heterogeneous expression of GFP-IQSEC1V2 and the image is saturated in more highly expressing cells. Accounting for this expression heterogeneity, the localization of V2 looks similar to V4.

We updated Fig 2G with a less intensity-saturated picture for GFP-IQSEC1 v2 in 3D. This shows in 3D a combination of cytoplasmic, cortical and nuclear labelling of GFP-IQSEC1 v2, which contrasts to GFP-IQSEC1 v4 that displays cortical labelling that is stronger than cytoplasmic intensity, as well as low nuclear levels. This aligns completely with the quantitation of these GFP-IQSEC1 variants in 2D.

We believe that the cytoplasmic labelling is exaggerated due to overexpression and potentially masks endosomal IQSEC1 v2. Indeed, antibodies to endogenous IQSEC1 v2 clearly label tubulovesicular compartments (Fig 2H-K, S2J,K).

3. The quantification of phospholipid localization is not very rigorous, because use of the PH-domain biosensors to quantify relative PIP levels requires the ability to measure only membrane-localized PH domains. It is not clear to me exactly how these measurements were performed, but simply using phalloidin staining to identify the cortex is not sufficient because it does not exclude PH probe that is proximal to the actin cytoskeleton but is not membrane-localized. Depending on the exact resolution of the microscope used, there could be significant overlap between a phalloidin image and a PH probe image that is not actually labeling membrane-localized PH probe. This is especially important given that the authors are comparing different cells populations that may have different membrane morphologies that would perturb such measurements.

We apologise for lack of clarity in our methods in the previous submission. We have updated the methods section (*‘2D and 3D Phenotypic analysis of fixed samples’*) to make our analysis clearer.

We counter that our approach is rigorous and robust. We use high-resolution, high content microscopy to sample thousands of cells stably expressing ultra-bright (mNeonGreen-tagged) fluorescent reporters for each phosphoinositide per condition. By co-staining with nuclear marker (Hoescht), F-actin (phalloidin), and cytoplasm marker (CellMask) we segment nuclear, cytoplasm, and cortex regions. This allows us to get a singular cytoplasm/membrane ratio per cell, that takes into account heterogeneity of cell shape and variability in expression of the reporter between cells. We previously reported that such an approach using high-content imaging and subcellular segmentation allows for exquisitely sensitive detection of changes in cortical (apical versus basal) localisation of phospholipids upon pharmacological inhibition of PIP enzymes (PMID: 30487552).

Whether validated that whether we designate the cell cortex using F-actin labelling (Phalloidin) or the periphery surrounding the cytoplasm (Cell Mask) gives the same result for PH domain levels at the cell cortex upon IQSEC1 inhibition (now Fig S7E). Moreover, we demonstrate that IQSEC1 effects are at the same magnitude as the strongest known inhibitor of PIP3 pathways: pan-inhibition of Class I PI3-kinases (new Fig S7F). If the reviewer prefers, we are happy to change our phrasing from 'cortical' to 'peripheral' to mitigate against the possibility proposed by the reviewer.

4. The conclusion that specific pools of Arf5/6 are critical seems to be based mostly on the result that expression of the fast cycling mutants does not rescue IQSEC depletion. Do the fast cycling mutants increase the GTP loading, or is Arf5/6 saturated under these conditions? Is there a constitutively active mutant that can be tested?

Our conclusion that a critical pool of ARF5/6 functions downstream of IQSEC1 is based on strong lines of evidence, but none of which are because '*expression of the fast cycling mutants does not rescue IQSEC depletion*'. Rather, despite IQSEC1 loss not affecting global ARF5/6-GTP loading by biochemical pulldown, all lines of evidence point to a crucial IQSEC1-ARF5/6 link (in addition to IQSEC1 working with ARF5/6 in all reported literature):

- Only ARF5/6 dual knockdown blocks invasion in 3D to levels and phenotype resembling IQSEC1 loss (Fig 3A-C). Although not included in this manuscript ARF1, ARF3, ARF4 knockdown have the opposite phenotype, inducing invasion. Note, the ARF2 gene was lost in humans.
- Dual overexpression of ARF5/6 stimulates invasion to levels and phenotype resembling IQSEC1 v2 overexpression (Fig 3D-F). This can be completely halted by co-knocking down IQSEC1 (Fig 3D-F), indicating that ARF5/6 effects require IQSEC1.
- IQSEC1-ARF5/ARF6 show extensive colocalization in 2D and 3D (Fig 3G, H, J). Similar localisations not were not seen for ARF1-4 (unpublished observations).
- Recruitment of a sensor for ARF GTP-loading (GGA1 NGAT) to both ARF5 and ARF6 is attenuated by IQSEC1 knockdown or pharmacological inhibition (Fig 3K).

To clarify, fast-cycling ARF mutants will not increase GTP-loading total levels. Rather these mutations change how quickly hydrolysis occurs such that ARF can be re-loaded with GTP.

We also 'constitutively active' mutants of ARFs because this designation is misleading. In ARFs, these GTP-preferential mutants lock ARF in a partial function. This causes abnormal cellular structures such as vacuoles which interfere with cellular function. ARFs are somewhat different in their function from the traditional idea of GTPases, such as Ras, where GTP-bound is thought of as 'ON', and GDP-bound as 'OFF'. The entire transition from GTP-bound to GDP-bound is required, such as GTP-ARF-dependent recruitment of vesicle coats onto budding membranes, followed by ARF-hydrolysis-dependent completion of budding to form a vesicle. Therefore mutants such as ARF6 Q70L are not actually 'constitutively active' ARF function but rather cause an abnormal, incomplete state. Hence why fast-cycling mutants were utilised as a more appropriate experiment to look at enhanced ARF function. Even in this instance, we find that the fast-cycling mutants fail to enhance the function of the ARFs to the levels that WT ARF overexpression can in either 2D or 3D (Fig S4A-G).

5. Why is Akt ppn used as the readout of PI3K inhibitor? What is the positive control for PI3K inhibitors efficacy? The PH-domain biosensors may be a more precise readout of PI3K inhibitor activity.

This was a great suggestion. Although Akt S473 phosphorylation is used extensively in the literature, we now also use antibodies to PIP2 and PIP3 and show that these are responsive to PI3K activation (myr-PI3K) and that both the antibodies and PH domain biosensors are responsive to PI3K inhibitor (now Fig S7E-H).

6. How was the pS473 spot area measured in Figure S6M&N? It looks like from the images that the increase in spot area may be due to overlapping spots, possibly due to changes in cell morphology because of IQSEC knockdown. It's also not clear how the statistics were calculated. How many cells

were 5analysed? Were the statistics calculated based on each spot being an independent measurement? In the case of image analysis of subcellular structures, it is not permissible to consider each measurement to be statistically independent if they are all taken from the same cell, because one or two cells with a very strong phenotype measured many times will strongly bias the results.

We apologise again for lack of clarity (see updated methods '*2D and 3D Phenotypic analysis of fixed samples*').

Images were taken using high-resolution, high content microscopy to sample thousands of cells. Each cell was segmented into different regions. We clarify that the values for quantitation of pAkt 'spots' (as are all our high content approaches) are the average of all spots per cell *not a datapoint per spot*. This includes the mean cortical/total intensity per cell or the mean cortical/total spot area per cell. n=3 independent experiments with 4 replicates/condition/experiment, 2,765 – 2,837 cells/condition. p values were derived using one-way ANOVA. (data in Fig S8I-J).

7. How many mice were included in the cohort for Figure 7B? 7D?

Scr (control, 12 mice) or *IQSEC1* KD4 shRNA (10 mice), consistent with efforts to minimise animal use in research to no more than absolutely required. We update the figure legend for Figure 7 to make this easier to find (originally stated in methods).

What are the p values? The lack of error bars suggests insufficient power for this experiment.

We apologise for any confusion. We include the statistical test used in the corresponding figure legends for each panel of Figure 7.

The presence or absence of error bars reflects their statistical appropriateness for a given metric. Where p-values are present but error bars are absent represents the use of Chi-squared analyses to identify if the count of membership to two groups is different wherein the value is Boolean (i.e. yes or no; e.g. the incidence or tumour formation per mouse (Fig 7B), or the incidence of macrometastasis per mouse (Fig 7F)). In this instance error bars are inappropriate. In contrast, where there is a quantal value between groups (i.e. different sizes, volumes, count of metastases per mouse) this can be analysed with a non-parametric test of independent group distribution (Mann-Whitney test), whereby variation within that metric is appropriately summarised by the use of error bars representing standard deviation from the mean.

The zero size of many tumors in Panel E suggest that there is an engraftment defect due to IQSEC1 knockdown.

We apologise again for not making this clearer in the previous submission. Figure 7B indicates no significant difference between Control (scramble shRNA) and *IQSEC1* KD groups in prostate tumour incidence, arguing against an engraftment defect. The 'zero' size represents small tumours below the detection threshold of the monitoring method (ultrasound), which were nonetheless verified as present in mice at the time of harvesting.

Furthermore, the authors claim that tumor size suggests a proliferation defect, but I disagree with this inference. If the authors want to make any conclusion regarding proliferation in these xenograft experiments, then they will need to measure proliferation and not tumor size, which is affected by the balance between proliferation, apoptosis, and cell movement.

This was a great suggestion that helped clarify the data. We paired analysis of proliferation (Ki-67) and apoptosis (cleaved caspase 3) through histological labelling of primary tumours and lymph node metastases. We must concede that we are limited in sample size such that we can only analyse metastases that form, of which few do upon *IQSEC1* KD. Despite this, this showed that the primary defect is indeed whether metastases can form and not major differences between proliferation or apoptosis of the primary tumour or metastases (Fig 7I,J).

8. There is no evidence that the lack of macromets is due to invasion, as it could be due to survival or proliferation defects, both of which are required for metastatic seeding and outgrowth into macromets.

Please see response to point above. In addition, we provide extensive new clinical characterisation of *IQSEC1*-ARF5/6 in prostate cancer, and across the TCGA pan-cancer atlas. This demonstrates that *IQSEC1* elevation is

linked to poor outcome and multi-year lowered survival across tumour types. Moreover, IQSEC1 elevation is strongly associated with increased phosphoinositide signalling and metastasis clinically.

9. It is not clear that IQSEC1 specifically depletes ARF GTP-binding in protrusions. Although it's not clear to me how the % overlap analysis was performed, a simple decrease in GTP-loading due to IQSEC knockdown would cause a decrease in colocalization overall and not just in protrusion tips. Furthermore, there are regions of GTP-bound ARF in Figure 3J that are not in the protrusion tips.

We agree with the reviewer. We apologise if we gave the impression that we considered IQSEC1 only acts on ARF5/6 only in the protrusions. Although we see IQSEC1-ARF5/6 co-localisation most prominently in protrusions in 3D (Fig 3H, J) and close to the leading edge in 2D (Fig 3G), they are each located in a number of places in the cell. Indeed, our ARF-GTP sensor detects analyses all pools. We have edited the manuscript to reflect this point.

10. Recovery of the invasion phenotype due to expression of the myristoylated PI3Kbeta looks relatively mild although I can't tell for sure based on how the data is presented. Thus it seems plausible that other effects of IQSEC knockdown might be causing the invasion defect.

Our data indicates that although myr-PI3KB robustly induced invasion and growth, IQSEC1 KD completely blocks invasion, but not the Area effect (Fig 6G,H). This supports our proposed interplay between IQSEC1 and PI3KB.

Reviewer #3 (Remarks to the Author); expert in prostate cancer and mouse/organoids models:

Major and overlapping signaling pathways are activated in prostate cancer to drive pleiotropic effects on cancer phenotype, including the stimulation of growth and invasion. However, prostate cancer progression proceeds through temporally separated stages with growth dysregulation typically preceding invasion. This application addresses an unresolved fundamental question, how is this signaling decoded to elicit differential effects on growth and invasion at different times and places. The data presented suggest one such mechanism, expression of IQSEC1 (a GTPase exchange factor) transcript isoforms that restrict PI(3,4,5)P3 production to discrete domains resulting in invasion-driving cellular protrusions.

This is a detailed mechanistic study whose data link several steps in the proposed signaling cascade, making a compelling case that spatially restricted, RTK-driven, IQSEC1 dependent, PI metabolism generates cellular protrusions that drive cancer cell invasion. While the general concept that spatially restricted signaling is important for migration/invasion is not novel, the particular IQSEC1 dependent mechanism described is novel in prostate cancer. Another interesting finding is that IQSEC1 is important not only for local AKT activation that drives invasion, but it is also required for trafficking of pAKT to other regions (juxtannuclear) where it drives overall cell growth.

The limitations of the current manuscript largely revolve around significance. All the experiments rely on established prostate cancer cell lines, almost exclusively the PC3 cell line. The potential limitation here is the mechanism described may be idiosyncratic in the PC3 line. There is some validation using other cell lines in supplementary figure 7, but this is restricted to use of a pharmacological ARF6 inhibitor. The manuscript would be strengthened by validation of the IQSEC1 mediated mechanism (genetic, imaging) in additional prostate cancer cell lines. Even better would be validation in more physiological experimental models (patient derived organoids, xenografts, etc.).

Thank you for your positive description of our manuscript. Your suggestions have made our manuscript much improved. We expanded our analyses to show in 3D that DU145 spheroids +/- HGF (whose invasion was blocked using 'ARF6' inhibitor) was similarly perturbed by IQSEC1 KD (new Fig S10). Moreover, we report that highly invasive human breast cancer cells (MDA-MB-231), highly metastatic mouse pancreatic cancer cells driven by oncogenic K-ras and PTEN loss (Kras-G12D PTEN fl/+), and metastatic patient-derived human pancreatic cancer cells (TKCC-07) all show near complete loss of invasion in 3D culture upon IQSEC1 KD.

We provide extensive new clinical characterisation of IQSEC1-ARF5/6 in prostate cancer and across the TCGA pan-cancer atlas (Fig 8, Fig S11). This demonstrates that IQSEC1 elevation is linked to poor outcome and multi-year lowered survival across tumour types. Moreover, IQSEC1 elevation is strongly associated with increased phosphoinositide signalling and metastasis clinically.

Additionally, all the IQSEC1 isoforms drive cell growth, with some having a greater effect on invasion. Hence effects on growth and invasion have not been genetically separated cleanly. This makes it difficult to ascribe the effects of IQSEC1 knockdown on metastasis in figure 7 to the mechanism described, rather than to general effects on tumor cell growth. The hypothesis predicts it should be possible to generate PC3 cells that grow robustly but metastasize weakly by removing specifically the invasion promoting IQSEC1 isoform; this should be tested if feasible.

This would be fantastic (and one of the earliest things we wanted to do in this project!). After many years, we have been unable to get isoform-specific editing. We have never been able to generate CRISPR clones for IQSEC1 which will assume would be due to severe growth defects upon complete IQSEC1 loss. This approach is also confounded by the nature of IQSEC1 isoforms: the coding sequence of V2 is the 3'UTR of other non-invasive isoforms. This needs to be a goal of future separate studies, as we are yet to come up with an approach that works to do this.

What we did was a pan-isoform RNAi and stably add back individual isoforms. Using this approach we show IQSEC1 v2 as a clear inducer of invasion (Fig 2), and new data showed V2 gave modest effects on proliferation or apoptosis (S2E-H). Our updated analysis of xenografted tumours revealed that IQSEC1 depletion minimally affects proliferation and apoptosis in those tumours or metastases that do form, but strongly suppresses formation of metastases (Fig 7).

Our extensive clinical characterisation of IQSEC1 mirrored the effect on metastasis observed in mice, and revealed that elevated IQSEC1 is strongly associated with metastasis in prostate cancer patients (Fig 10). In addition, by comparing matched normal and tumour tissue from prostate cancer patients we identify an increase in IQSEC1 expression and similarly a switch from the non-invasive IQSEC1 v1 to the pro-invasive IQSEC1 v2. This switch to IQSEC1 v2 occurred in eight tumour types from the pan-cancer TCGA atlas.

Our results therefore now extensively point to a pro-metastatic role of IQSEC1 v2, which is clinically relevant to a number of tumour types.

Technical issues to be resolved:

1) I am not sure “spatially restricted” is the most accurate term to describe the mechanism since IQSEC1 is also important for trafficking signals elsewhere; is “spatially concentrated” signaling more accurate?

Our data indicate that IQSEC1 does not control general synthesis of phosphoinositides, but instead restricts them to the cell surface (See Reviewer 1, point 2). However, we take this point into consideration and have changed the title and corresponding statements in the manuscript to ‘spatially enriched’. We are happy with either the change to ‘enriched’ or to change back to ‘restricted’, considering the latter is supported by our new data.

2) The authors may want to speculate on what the spatially restricted signaling is doing to drive formation of protrusions. Is it local macromolecular synthesis?

This will require future studies to identify. We add the following statement to the discussion:

“The identity of invasion-specific PIP₃ effectors remains to be demonstrated, but one likely candidate is Rac1, which is dependent on PIP₃ for activation.”

3) The authors make the assumption that their morphological measure (i.e. area, compactness, protrusions) accurately predicts physiological invasion. This assumption should be validated in a manner more rigorous than wound healing assays (i.e. Boyden chamber assays, metastasis in vivo, etc.).

We indeed had validated that the compactness measure of invasion from spheroids represented invasion in an orthogonal assay, using wounded monolayers of cells embedded in matrix (Fig S1H). Here, IQSEC1 similarly blunts invasion. For the sake of an already dense manuscript we didn't further include the cross validation we had performed for several additional assays.

There is no evidence that a Boyden chamber assay is any more or less physiological than matrix-embedded wound healing assays or spheroid invasion. It could be argued that Boyden assays select for single cell invasion, whereas increasing evidence suggests collective movement as a major mechanism for metastasis. In the spheroid and scratch wound assays cells have the choice of collective or single cell movement. In the vast majority of cases we see collective movement, not single cells. This argues against Boyden chamber use.

We have performed the most robust assay we could use with these cells, xenografting into the prostate and looking for distal metastasis from the prostate. We believe this to be the most physiological assay for metastasis. Other experimental metastasis assays such as tail vein injections are traditionally proposed to assay 'metastasis to lung'. It could be argued that the latter approach is described as metastasis, but is more accurately an assay for metastatic seeding in a hyper-vascularised tissue rather than requiring *bona fide* tissue invasion that the intraprostatic xenografts we performed require.

4) The data presented in figure 2J showing highly localized IQSEC1v2 immunostaining in the protrusion tip is compelling, but there is no quantitative measure of what fraction of protrusions show this immunostaining pattern. This should be stated or shown. If other isoform specific antibodies are available, it would be important to compare them in this assay.

This is an excellent suggestion. We now include analysis with a pan-isoform antibody and IQSEC1 v2-specific antibody in Fig 2K. We added images showing spheroid with multiple protrusions stained with these antibodies (now S2 F and G). Unfortunately, we have no other isoform-specific antibodies that work well in 3D.

5) It is not clear why the clinical correlations presented in figure 1J utilize a TCGA dataset (n=487) while the clinical correlations presented in figure 7M use a different, smaller dataset (n=79). Why were these particular data sets chosen? The manuscript would be strengthened by analysis of multiple, independent publically available datasets inform the reader how general the findings are.

We thank you very much for this suggestion as it spurred the much larger analysis we present in this revision. We replace these smaller cohorts with extensive analysis of the TCGA pan-cancer atlas, and multiple cohorts of prostate cancer patients. As described in answer to your first point, this show extensive association of IQSEC1 with metastasis and poor clinical outcome across over 10,000 patients and many cancer types.

6) The authors may want to comment on whether this mechanism is unique to prostate cancer or can be generalized to other cancers.

Please see answer to your queries above.

REVIEWER COMMENTS

Reviewer #1 (Remarks to the Author):

The authors have done a good job of addressing my earlier concerns. While many of the effects of IQSec1 knockdown are subtle, the authors have convinced me that these subtle differences are indeed physiologically important, and are largely due to redistribution of phosphoinositides rather than changes in cellular levels. The overall effect of IQSec1 knockdown is clearly important, given the virtually complete inhibition of metastasis in the in vivo model. My concerns were more related to mechanism, and these have been largely addressed.

Reviewer #2 (Remarks to the Author):

I would like to thank the authors for their efforts put into their revision. They have addressed several of the reviewers' points and the manuscript has overall improved because of this. However, one critical concern remains, which they did not adequately address in their rebuttal or via changes to the manuscript. This involves the claim, which is very prominent throughout the manuscript including the title and abstract, that the specific location of phosphoinositide metabolism regulates metastasis. Although they have some circumstantial evidence suggesting that IQSEC1 may alter the spatial localization of phosphoinositides, these results are, as also pointed out by Reviewer 1, the weakest part of this otherwise impressive effort. As I describe below, there are very serious concerns about the rigor of the imaging and analysis, which were not addressed, and therefore their claims regarding the spatial distribution of phospholipids must be removed, in particular because they are so prominent. Reviewer 1 also suggested that they modify these claims, but they have not. For example, here are two direct quotes from the abstract, neither of which are justified based on the data:

"In contrast, select pro-invasive IQSEC1 variants restrict PI(3,4,5)P3 production to discrete cortical domains to form invasion-driving protrusions."

"Spatial enrichment of phosphoinositide metabolism therefore is a switch to induce invasion over growth in response to the same external signal."

My reasons for this position are outlined below. These concerns may seem pedantic but as we move forward by incorporating image-based measurements into biological studies it is important that we do not set a standard whereby results can be misinterpreted or biased by technological issues. All of this being said, I am sympathetic to the authors desire to publish their work and I do not want to prevent them from doing so. Aside from the phospholipid analysis, they have performed a lot of very beautiful, rigorous work in this manuscript. I suggest that the authors remove their claims regarding phosphoinositide enrichment manuscript. If they are willing to do this, then I see no reason why the manuscript cannot be published with minimal additional work.

On an unrelated but also critical point, the authors need to research and cite the relevant literature more fully and more carefully. There is an extensive literature on the role of PI3K in cell protrusions, which they have completely neglected. Furthermore, their claim (in their rebuttal) that "this is the first demonstration that cells coordinate this polarity across a multicellular scale to promote collective cell invasion" is false in two ways. First, they do not show that cells coordinate this polarity across a multicellular scale, only that cells do it in multicellular organoids. Second, I know of at least one prior publication (PMID: 29634937) showing that cells in 3D organoids show PIP3 enrichment at

the cell protrusion. Additionally, their statement “ARF6 can activate PI3K signalling in melanoma57” is incomplete. In the study by Yoo et al. they show not only that ARF6 can activate PI3K, but that this increased PI3K signaling shows enhanced distribution to cellular protrusions, which is also what the authors of this study claim, and this must be noted to make an accurate citation of published work. It is completely unacceptable to neglect the published literature in this way.

As it currently stands, the data presented in Figure 6C and Figure S7B and S7C, which are the only quantitative data on the topic, are inaccurate measures of phospholipid abundance because the imaging was not performed appropriately and this study lacks appropriate controls for the approach. Furthermore, as far as I can tell, their analysis does not address the issue of if phospholipids are enriched in protrusions, although this is what they are claiming.

The authors’ response and changes to the manuscript in response to my concern over the microscopy approach and subsequent quantification of phospholipid abundance were insufficient to evaluate their biosensor quantification approach. It seems that this information is only available in their previous publication. Based on this publication, it seems that a single confocal slice was used to quantify biosensor localization and thus phospholipid abundance. First, using confocal microscopy to quantify localization of the PH probes is not a good way to quantify their localization because of the inaccuracy in measuring membrane proximal fluorescence. These PH probes were previously developed and evaluated using TIRF microscopy, which is able to report membrane proximal fluorescence within 200 nm, but it is not clear how accurate these probes are when imaged using confocal microscopy. If the authors wish to maintain their claims regarding phospholipid abundance, then they need to measure and report the axial and lateral resolution of their microscope using the settings with which they acquired the data and show that this microscopy modality can accurately report changes in phospholipid metabolism. Otherwise, there is no way to evaluate the accuracy of their approach. This information is critical because inaccuracy in image acquisition can render their measurements sensitive to changes in membrane topology. If all cells under all conditions had the same shape and membrane topology then this would be less of a concern, but it is possible that perturbed cells simply have a different membrane topology and this causes a shift in their measurements of phospholipid abundance. Their observation that cell shape has a dramatic effect on their measurements actually strongly suggests the possibility that their measurements are simply due to morphological differences. Their observation that IQSEC1 knockdown results in similar shifts to their measures of both PIP2 and PIP3 abundance also suggest a systemic shift in their measure may be present.

Another concern is that a single confocal slice is not sufficient to capture the abundance of biosensor in morphologically different cells. For example, a flat cell imaged near the coverslip via confocal microscopy would most likely contain fluorescence from both the dorsal and ventral membrane as well as the cytosol in between. A less well spread cell imaged similarly would only show the ventral membrane and cytosol. Furthermore and perhaps more importantly, cells in 3D culture exhibit tortuous membrane protrusions and a single confocal slice is not sufficient to measure overall biosensor abundance. Thus, we have no way of knowing if the single image slice apparently used for this study is unbiased or representative of the whole cell or even a certain area of interest. If the authors wish to maintain their claims regarding phospholipid abundance, then they need to prove that the imaging was performed in a systemic, unbiased way that is equally representative for all conditions.

Equally important, the image analysis as they have reported it is completely a black box and we have no way to judge its accuracy. This point is critical because the segmentation of the cell outline, used to quantify cortically proximal fluorescence, will dramatically affect the cortical/total fluorescence ratio. For example, if a cell outline has many small invaginations that are blurred by the imaging or segmentation, then the apparent cell edge is not the true cell edge and their measurements will not

accurately report membrane proximal biosensor. Again, such membrane topologies are likely to be systematically affected by perturbations such as IQSEC1 knockdown. The observation that IQSEC1 knockdown decreases their measures of both PIP2 and PIP3 would be consistent with a morphological shift that affects their measure. If the authors wish to maintain their claims regarding phospholipid abundance, then they need to report control experiments demonstrating that their pipeline shows no effect of IQSEC1 knockdown on a membrane binding probe that is unaffected by IQSEC1.

Reviewer #3 (Remarks to the Author):

This manuscript addresses the fundamental question how signaling pathways elicit differential effects on growth and invasion. The data presented suggest one such mechanism involves expression of IQSEC1 transcript isoforms that favor PI(3,4,5)P3 production at discrete subcellular locations to drive cell invasion.

The strengths of the original manuscript remain in this revised version. It describes a detailed mechanistic study linking several steps in the proposed signaling cascade to make a compelling case that spatially enriched, IQSEC1 dependent, PI metabolism generates cellular protrusions that drive cancer cell invasion. Data is abundant, and largely convincing. Significance of the study has been increased by new figure 8 which examines public prostate cancer data derived from human clinical specimens to demonstrate a correlation between the proposed mechanism and patient outcome measures. Further, the figure extends analysis to additional cancers to suggest the mechanism may more generally participate in driving poor patient outcomes. Rigor has also been increased by analysis of additional experimental cancer models from both prostate (DU145) and additional cancer types (breast, pancreatic) in new supplementary figure 10.

Most of the technical issues I raised have been addressed adequately. While not all points have been addressed as originally envisioned, the authors have added additional experiments where feasible that touch on these points. For example, the pan-isoform knockdown and add back experiments performed do address genetic separation of growth promoting and invasion promoting functions as predicted by the hypothesis. The fraction of protrusions showing relevant immunostaining patterns has now been quantitated. For those issues that have not been addressed experimentally, the authors present a reasonable rebuttal.

From the perspective of my original review, therefore, the revised manuscript has addressed noted deficiencies and is suitable for publication.

Response to Reviewer 2.

We thank the reviewer for their positive assessment of much of our manuscript. We address below the reviewer's continued concerns regarding our imaging approach and our conclusion that IQSEC1 contributes to phospholipid localisation. We believe that all of the points below can be addressed through clarification of data that was already in the manuscript or that we had performed as background validation.

Reviewer #2

I would like to thank the authors for their efforts put into their revision. They have addressed several of the reviewers' points and the manuscript has overall improved because of this. However, one critical concern remains, which they did not adequately address in their rebuttal or via changes to the manuscript. This involves the claim, which is very prominent throughout the manuscript including the title and abstract, that the specific location of phosphoinositide metabolism regulates metastasis. Although they have some circumstantial evidence suggesting that IQSEC1 may alter the spatial localization of phosphoinositides, these results are, as also pointed out by Reviewer 1, the weakest part of this otherwise impressive effort. As I describe below, there are very serious concerns about the rigor of the imaging and analysis, which were not addressed, and therefore their claims regarding the spatial distribution of phospholipids must be removed, in particular because they are so prominent. Reviewer 1 also suggested that they modify these claims, but they have not.

For example, here are two direct quotes from the abstract, neither of which are justified based on the data:

"In contrast, select pro-invasive IQSEC1 variants restrict PI(3,4,5)P3 production to discrete cortical domains to form invasion-driving protrusions."

"Spatial enrichment of phosphoinositide metabolism therefore is a switch to induce invasion over growth in response to the same external signal."

My reasons for this position are outlined below. These concerns may seem pedantic but as we move forward by incorporating image-based measurements into biological studies it is important that we do not set a standard whereby results can be misinterpreted or biased by technological issues. All of this being said, I am sympathetic to the authors desire to publish their work and I do not want to prevent them from doing so. Aside from the phospholipid analysis, they have performed a lot of very beautiful, rigorous work in this manuscript. I suggest that the authors remove their claims regarding phosphoinositide enrichment manuscript. If they are willing to do this, then I see no reason why the manuscript cannot be published with minimal additional work.

We respond only to Reviewer 2, as I report that Reviewer 1 states that our changed addressed their previous concerns. We hope that Reviewer 2 will see that our response to their points below further buttresses support for the technical accuracy of our approach and our assertion of an effect on phosphoinositide localisation. We do take the point of being accurate about the description of the data. We have therefore softened the tone of our statements regarding phosphoinositide localisation in protrusions to being at the cell periphery throughout the manuscript (indicated in blue in manuscript).

On an unrelated but also critical point, the authors need to research and cite the relevant literature more fully and more carefully. There is an extensive literature on the role of PI3K in cell protrusions, which they have completely neglected. Furthermore, their claim (in their rebuttal) that "this is the first demonstration that cells coordinate this polarity across a multicellular scale to promote collective cell invasion" is false in two ways. First, they do not show that cells coordinate this polarity across a multicellular scale, only that cells do it in multicellular organoids. Second, I know of at least one prior publication (PMID: 29634937) showing that cells in 3D organoids show PIP3 enrichment at the cell protrusion. Additionally, their statement "ARF6 can activate PI3K signalling in melanoma57" is incomplete. In the study by Yoo et al. they show not only that ARF6 can activate PI3K, but that this increased PI3K signaling shows enhanced distribution to cellular protrusions, which is also what the authors of this study claim, and this must be noted to make an accurate citation of published work. It is completely unacceptable to neglect the published literature in this way.

Any lack of description of literature was unintentional and due to trying to provide balanced discussion to the large breadth of fields we work across in this manuscript. We correct this by adding the following statement with additional references into the discussion (p21, in blue in the manuscript), with a number of additional citations (including those above):

“The role of phosphoinositides, particularly PIP_3 , in regulating leading edge formation or protrusion activity has been extensively reported⁶⁰⁻⁶⁷, including $ARF6$ being a driver of $PI3K$ enrichment in melanoma cell protrusions⁵⁷. Our data further supports central roles for ARF GTPases in invasion.”

Reviewer Figure 1. Enrichment of the $PI(3,4,5)P_3$ probe mNeonGreen-GRP1 in protrusions. Pseudocolouring of probe intensity using the Fire LUT. **A.** Maximum intensity projection of all Z-slices (1-123), slice 60, slice 82, or Max projection of slices (70-105). **B.** Linescan analysis of PIP_3 probe intensity in cyst body or protrusion in the protrusion in slice 60. **C.** Individual representative images of enrichment of PIP_3 probe in the protrusion occurring in slice 70-105. **D.** Linescan analysis of PIP_3 probe intensity in cyst body or protrusion in the protrusion occurring through slice 70-105. Arrowheads, PIP_3 in protrusions.

As it currently stands, the data presented in Figure 6C and Figure S7B and S7C, which are the only quantitative data on the topic, are inaccurate measures of phospholipid abundance because the imaging was not performed appropriately and this study lacks appropriate controls for the approach. Furthermore, as far as I can tell, their analysis does not address the issue of if phospholipids are enriched in protrusions, although this is what they are claiming.

We assert that our imaging approach is accurate, robust, controlled, and supported by uncommonly deep sampling of cells (thousands per condition). We apologise if we did not convey this well enough previously.

Are phospholipids enriched in protrusions?

We provide as Reviewer Figure 1 an expanded view of the PH-GRP1 $PI(3,4,5)P_3$ reporter in a 3D acinus demonstrating PIP_3 enrichment in the invasive protrusions. This was taken using super-resolution confocal imaging

of the PI(3,4,5)P3 probe (mNeonGreen-tagged PH-GRP1) obtained on a Zeiss 880 LSM with airyscan capability with a Plan-Apochromat 63x/1.4 Oil DIC M27 lens, with a Z plane step size of 0.385 μm .

A challenge in displaying this data is an acinus can have multiple protrusions and because of the 3-Dimensional nature of the acinus those protrusions can occur across multiple Z planes. We provide quantitation of both a) PIP3 in the protrusion and in cell body in the same section, and b) across multiple sections of the same protrusion using line scan analysis (Reviewer Figure 1). In this, there is clear enrichment of PIP3 signal in protrusions compared to the cell body cortex.

A challenge is how to display this enrichment in a simple 3D image. In this example, we took 123 slices to cover the entire acinus. Maximum projection of the entire acinus depicts where protrusion occur, but biases fluorescence to slices at the top of the acinus and fails to clearly indicate the enrichment of PIP3 in the protrusion (Reviewer Figure 1A, top panel). In contrast, single slices clearly show PIP3 enrichment in protrusions but fail to give a sense of how this relates to enrichment specifically in the tips versus in the rest of the acinus (which is part of the reviewer's point) (Reviewer Figure 1A, rows 2-4). Presentation of all of these approaches in this Reviewer Figure 1 gives a clearer sense of the protrusion enrichment of PIP3. For the sake of space, we cannot provide such an expanded figure set in the main manuscript. We therefore reasoned that a Maximum Intensity Projection of a focal region of the acinus including both the protrusion and some of the acinus body best represented this protrusion tip enrichment profile, particularly when represented by intensity conveyed using an intensity-based pseudocolouring look-up table. This is what was presented in Figure 6B of the manuscript. We are open to including this figure as an additional supplementary figure if the reviewer deems necessary.

The authors' response and changes to the manuscript in response to my concern over the microscopy approach and subsequent quantification of phospholipid abundance were insufficient to evaluate their biosensor quantification approach. It seems that this information is only available in their previous publication.

We have now extensively updated the methods section to further elaborate our imaging approach methodology.

Based on this publication, it seems that a single confocal slice was used to quantify biosensor localization and thus phospholipid abundance. First, using confocal microscopy to quantify localization of the PH probes is not a good way to quantify their localization because of the inaccuracy in measuring membrane proximal fluorescence. These PH probes were previously developed and evaluated using TIRF microscopy, which is able to report membrane proximal fluorescence within 200 nm, but it is not clear how accurate these probes are when imaged using confocal microscopy.

The advantage of TIRF microscopy described above is also its disadvantage. It allows examination of coverslip proximal fluorescence of 2D grown cells. This biases analyses to the basal membrane-proximal region and ignores the possibility that a major PIP localisation or function could occur outside that region. In more complex 3D collections of cells, indeed other regions of the cell (eg apical domain) have extremely different PIP dynamics to the basal surface of 2D cells. Accordingly, these probes have been extensively utilised in PIP cell biology for many years, much of which has not been with TIRF microscopy. This includes *in vivo* in mice, in flies, worms, yeast, Dictyostelium and by us and others in mammalian 3D culture – all of which was performed by confocal microscopy. TIRF is not the sole use historically for these probes and would therefore be inappropriate for analysis of 3D culture.

If the authors wish to maintain their claims regarding phospholipid abundance, then they need to measure and report the axial and lateral resolution of their microscope using the settings with which they acquired the data and show that this microscopy modality can accurately report changes in phospholipid metabolism. Otherwise, there is no way to evaluate the accuracy of their approach.

We update the methods to include the requested information on our microscopy. We have previously shown (Roman-Fernandez, Nat Comms, 2018; PMID: 30487552) using these PIP reporters by confocal microscopy in 3D culture that they are indeed responsive to changes in PIP metabolism, and that we can detect this using the same methods we use here. Particularly, we showed that pharmacological inhibition of the PIP3 metabolising 5-phosphatase SHIP1, which is localised the basolateral membrane in polarised cysts with a lumen, increased PIP3 levels specifically at the basolateral membrane (using the same mNeonGreen-PH-GRP1 reporter used here) whilst not affecting the levels of any of the other PIP reporters that we examined (eg. PI(4)P2, PI(3,4)P2, PI(4,5)P2). This reveals exquisite, polarised domain-localised detection of PIP metabolism in a complex 3D culture using these probes and confocal microscopy.

These probes have also been extensively utilised in the literature by other groups to examine changes in PIP metabolism by confocal. An example in 2D cells is (PMID: 30591513, Fig 1B-C), and in 3D in the work of Andy

Ewald's group that the reviewer indicated above detailed PIP3 in protrusions in organoids (PMID: 29634937, Fig 1E).

This information is critical because inaccuracy in image acquisition can render their measurements sensitive to changes in membrane topology. If all cells under all conditions had the same shape and membrane topology then this would be less of a concern, but it is possible that perturbed cells simply have a different membrane topology and this causes a shift in their measurements of phospholipid abundance. Their observation that cell shape has a dramatic effect on their measurements actually strongly suggests the possibility that their measurements are simply due to morphological differences. Their observation that IQSEC1 knockdown results in similar shifts to their measures of both PIP2 and PIP3 abundance also suggest a systemic shift in their measure may be present.

The reviewer may have missed that we already corrected for the possibility that shape could be affecting our analysis in our previous submission. Figure 6C indicated that IQSEC1 depletion resulted in decreased PI(4,5)P2 and PI(3,4,5)P3 peripheral-to-total intensity across all cells. However, in Figure S7B-C we classified cells into the three major shape categories observed (Spindle, Spread, Round) and even within these same shape categories the cortical levels of PI(4,5)P2 and PI(3,4,5)P2 were decreased upon IQSEC1 depletion. These verify that irrespective of shape, IQSEC1 depletion depletes peripheral PI(4,5)P2 and PI(3,4,5)P2 levels.

We also analysed PI(4)P cortical levels using a similar approach and the PI(4)P probe mNeonGreen-P4M-SidM. Critically, surface levels of PI(4)P were not significantly altered by IQSEC1 depletion. We did not include this in the previous submission due to space, but here include this as Reviewer Figure 2. Given that ARF-PI4P5K module described here should act at this exact point (after PI(4)P to control PI(4,5)P2 and PIP3 levels), this supports our assertion of a specific effect on this stage of cortical PIP metabolism rather than a general effect on the cell surface or a systemic bias based on cell shape.

Another concern is that a single confocal slice is not sufficient to capture the abundance of biosensor in morphologically different cells. For example, a flat cell imaged near the coverslip via confocal microscopy would most likely contain fluorescence from both the dorsal and ventral membrane as well as the cytosol in between. A less well spread cell imaged similarly would only show the ventral membrane and cytosol. Furthermore and perhaps more importantly, cells in 3D culture exhibit tortuous membrane protrusions and a single confocal slice is not sufficient to measure overall biosensor abundance. Thus, we have no way of knowing if the single image slice apparently used for this study is unbiased or representative of the whole cell or even a certain area of interest. If the authors wish to maintain their claims regarding phospholipid abundance, then they need to prove that the imaging was performed in a systemic, unbiased way that is equally representative for all conditions.

Please see response above for correction against shape inducing bias. As part of our validation in this approach, we had examined how imaging modalities affected our data. We update this in the methods (in blue text in the manuscript), and present the results as Reviewer Figures 3 and 4. The summary is: whether we scan at 63x or 20x, analyse a single plan, a maximum projection of multiple planes, or take the average of multiple plans, we obtain the corroborative results.

To ensure robust sampling of phenotypes, we used a Perkin Elmer Opera Phenix automated high-content confocal microscope. This allows us to automate imaging of thousands of cells per condition (in some instances over 10,000 cells sampled per condition). Cells are first detected via an in-built algorithm that detects at low (5x or 10x) magnification where cells are present in the well, and then rescans at higher magnification (PreciScan Intelligent Acquisition function). We performed the higher magnification scans with a) a 20x Air, 0.4 NA objective with Two-peak autofocus and binning factor of 2 or b) 63x water, 1.15NA, two-peak autofocus, binning factor of 2 modalities. To ensure that accurate sampling occurs in cells (as cells may have differing heights) we collected three Z planes per field of imaging on each lens, wherein at 20x this represented a 2µm step size or at 63x a 1µm step size.

Each plane was either individually subjected to image analysis or the 3 planes were maximum-projected before image analysis. For selection of a single plane, post analysis we automated selection of the plane with the largest cell area using KNIME data analytics software. In Reviewer Figure 3, the effect of IQSEC1 depletion on PI(4,5)P2 surface levels was observed whether cells were imaged at 63x or 20x, whether a single slice was chosen, whether we took the average of the three slices, or whether we analysed maximum intensity projections of all three sliced. This validation supports that a single slice was indeed sufficient to capture representative cortical PIP levels across cells and the effect of IQSEC1 depletion on peripheral PIP levels.

Equally important, the image analysis as they have reported it is completely a black box and we have no way to judge its accuracy. This point is critical because the segmentation of the cell outline, used to quantify cortically proximal fluorescence, will dramatically affect the cortical/total fluorescence ratio. For example, if a cell outline has many small invaginations that are blurred by the imaging or segmentation, then the apparent cell edge is not the true cell edge and their measurements will not accurately report membrane proximal biosensor.

We use automation for cell segmentation. We update the methods to be explicit about our approach. Briefly, we first anchor the position of single cells by detection of the nuclei. We expand to find the cell area using the cytoplasm surrounding each nucleus as delineated by detection of the cytoplasm labelling Whole Cell Stain. All cells touching the image border are removed. Cells are measured for mNeonGreen intensity, to ensure that subsequent analysis only occurs on cells expressing the PIP probe. Using machine learning approaches, cells are user-classified into 3 shapes (Spindle, Round, Spread). We delineated and compared detection of the cell periphery by alternate means: a) detected via either the presence of F-actin labelled by Phalloidin at the edge of the cytoplasm region or b) by the lack of Whole Cell Stain labelling. The 'cell membrane' region is defined as Outer border: -2 px (-1.194 μm), Inner border: 5 px (2.985 μm). No blurring or modification of signal functions are applied at this region. As indicated in the previous response, either method for membrane section results in the same findings.

What results as the 'membrane' region is segmentation of $\sim 4 \mu\text{m}$ around the cell border, likely close to but not strictly the cell surface (vesicles closely apposed to the surface, would not be distinguished from the bona fide plasma membrane). To highlight this point, we have altered our statements of 'cortical' to 'peripheral' wherein we clearly define on p14 (highlighted in blue) that, "*We examined the ratio of lipid biosensor at the cell periphery, which includes a region of $\sim 4 \mu\text{m}$ at or very close to the cell surface, to total expression.*"

Again, such membrane topologies are likely to be systematically affected by perturbations such as IQSEC1 knockdown. The observation that IQSEC1 knockdown decreases their measures of both PIP2 and PIP3 would be consistent with a morphological shift that affects their measure. If the authors wish to maintain their claims regarding phospholipid abundance, then they need to report control experiments demonstrating that their pipeline shows no effect of IQSEC1 knockdown on a membrane binding probe that is unaffected by IQSEC1.

See response above about correction for cell shape. Also see above Reviewer Figure 2 which shows that the probe for PI(4)P is not affected by IQSEC1 depletion, allowing us to main our claim of a specific effect of IQSEC1 on PI(4,5)P2 and PI(3,4,5)P3.

One-Way ANOVA. No matching or pairing.
 Assume Gaussian distribution (use ANOVA).
 Tukey: P-value style: * ≤ 0.05 , ** ≤ 0.005 , *** ≤ 0.0005 , **** ≤ 0.0001

Reviewer Figure 3. Validation of detection methodologies. The effect of IQSEC1 depletion lowering peripheral PI(4,5)P2 or PI(3,4,5)P3 levels is robustly detected irrespective of imaging at 63 x without correcting for shape (left), 20x without correcting for shape (second column), 20x correcting for shape and normalising to each scramble (third column), or 20x correcting for shape and normalising to Round scramble values (right). Statistical methodology indicated at bottom left of figure.

PIP2_Periphery/Total

PIP3_Periphery/Total

Reviewer Figure 4. Validation of Z-plane detection methodologies. The effect of IQSEC1 depletion lowering peripheral PI(4,5)P2 or PI(3,4,5)P3 levels is robustly detected irrespective of whether analysis is performed on a) a single plane (cell with largest area selected from each stack), b) the average of 3 planes, or c) upon Maximum intensity projection of all planes, with or without correction for shape classification.

REVIEWER COMMENTS

Reviewer #2 (Remarks to the Author):

After reading the responses to my concerns over imaging and image analysis, it is clear that the authors do not understand why their approach is not quantitatively rigorous. Despite their claim that the imaging is "super resolution", it is not super resolution and in particular, the axial resolution is insufficient to resolve the complexities of membrane topology that exist in 3D cellular structures. Their analytical workflow is also insufficient to support their claims. This is perfectly exemplified by their attempt to illustrate the difficulties of analyzing 3D data in the Reviewer Figure 1, which is not at all convincing. This really needs to be done on a 3D surface and not on 3D data as slices, and it needs to be done on data that is acquired with sufficient z-sampling and resolution. Furthermore, it is not relevant whether previous researchers may have used these probes in a qualitative fashion in 3D data. Here, the authors claim quantitative rigor and I strongly disagree with that assertion.

Their claim that stratifying cells according to shape, and showing that all shapes respond similarly also illustrates a fundamental misunderstanding of the 3D analytical challenges this work faces. The issue is not completely CELL SHAPE, but membrane microtopology, which is not addressed at all by their analysis of cell shape.

Finally, the changes they made to the manuscript are superficial and do not address these fundamental limitations of their analytical workflow. For example, their chosen terms, spatial enrichment at the cell periphery, is meaningless because all membrane lipids are enriched in the cell membrane. They have also kept much of their original language regarding spatial enrichment and protrusion in the manuscript, which is not justified. Most likely what they are seeing is a global shift in phospholipid metabolism, which has very little to do with spatial properties at all.

Reviewer #2 (Remarks to the Author):

After reading the responses to my concerns over imaging and image analysis, it is clear that the authors do not understand why their approach is not quantitatively rigorous. Despite their claim that the imaging is "super resolution", it is not super resolution and in particular, the axial resolution is insufficient to resolve the complexities of membrane topology that exist in 3D cellular structures. Their analytical workflow is also insufficient to support their claims. This is perfectly exemplified by their attempt to illustrate the difficulties of analyzing 3D data in the Reviewer Figure 1, which is not at all convincing. This really needs to be done on a 3D surface and not on 3D data as slices, and it needs to be done on data that is acquired with sufficient z-sampling and resolution. Furthermore, it is not relevant whether previous researchers may have used these probes in a qualitative fashion in 3D data. Here, the authors claim quantitative rigor and I strongly disagree with that assertion.

Their claim that stratifying cells according to shape, and showing that all shapes respond similarly also illustrates a fundamental misunderstanding of the 3D analytical challenges this work faces. The issue is not completely CELL SHAPE, but membrane microtopology, which is not addressed at all by their analysis of cell shape.

Finally, the changes they made to the manuscript are superficial and do not address these fundamental limitations of their analytical workflow. For example, their chosen terms, spatial enrichment at the cell periphery, is meaningless because all membrane lipids are enriched in the cell membrane. They have also kept much of their original language regarding spatial enrichment and protrusion in the manuscript, which is not justified. Most likely what they are seeing is a global shift in phospholipid metabolism, which has very little to do with spatial properties at all.

We have substantially edited the manuscript to tone down all assertions of spatial regulation of phosphoinositide metabolism as requested by Reviewer 2. Changes are indicated throughout the manuscript in blue text. This includes changing the title to "An ARF GTPase module promoting invasion and metastasis through regulating phosphoinositide metabolism". The main results clearly and explicitly state the limitations of our approach, and we further emphasise this with additional text in the discussion section, including future avenues to expand on this work. We have also moved experimental data relating to these conclusions from the main manuscript to the supplementary data section.